# EFFICIENT ALGORITHMS FOR INCREMENTAL METRIC BIPARTITE MATCHING

**Ritesh Seth**[1]    **Mrinal Garg**[2]    **Sujoy Bhore**[2]    **Sharath Raghvendra**[3]    **Syamantak Das**[1]

[1]IIIT Delhi                    [2]IIT Bombay                    [3]NC State University

## ABSTRACT

The minimum-cost bipartite matching between two sets of points $R$ and $S$ in a metric space has a wide range of applications in machine learning, computer vision, and logistics. For instance, it can be used to estimate the 1-Wasserstein distance between continuous probability distributions and for efficiently matching requests to servers while minimizing cost. However, the computational cost of determining the minimum-cost matching for general metrics spaces, poses a significant challenge, particularly in dynamic settings where points arrive over time and each update requires re-executing the algorithm. In this paper, given a fixed set $S$, we describe a deterministic algorithm that maintains, after $i$ additions to $R$, an $O(1/\delta^{0.631})$-approximate minimum-cost matching of cardinality $i$ between sets $R$ and $S$ in any metric space, with an amortized insertion time of $\widetilde{O}(n^{1+\delta})$ for adding points in $R$. To the best of our knowledge, this is the first algorithm for incremental minimum-cost matching that applies to arbitrary metric spaces.

Interestingly, an important subroutine of our algorithm lends itself to efficient parallelization. We provide both a CPU implementation and a GPU implementation that leverages parallelism. Extensive experiments on both synthetic and real world datasets showcase that our algorithm either matches or outperforms all benchmarks in terms of speed while significantly improving upon the accuracy.

## 1 INTRODUCTION

Large-scale online logistics systems typically consist of a fixed fleet of vehicles or robots, while service requests appear dynamically over time. The task is to maintain a cost-effective assignment of requests to servers. For example, the New York Taxi System processes more than 300,000 ride requests daily with a fleet of several thousand taxis NYC Taxi & Limousine Commission (2024). A plethora of prior studies model this problem as the *classical minimum cost bipartite matching problem* (Tong et al., 2016; Zhao et al., 2019; Ke et al., 2019; Tong et al., 2023; Qin et al., 2021; Abeywickrama et al., 2021). A non-trivial challenge in adapting minimum cost matching to a dynamic framework is that recomputing a matching from scratch whenever a new request arrives is computationally prohibitive and slows down downstream decisions: if each assignment depends on a full recomputation, the time to compute may exceed the time to dispatch a taxi, significantly increasing passenger wait times. When requests arrive faster than the system can process them, queues build up, causing cascading delays and further degrading responsiveness. This setting raises two key challenges: *(i)* can we design data structures that maintain an approximate minimum-cost matching while supporting efficient insertions, and *(ii)* can these structures process new arrivals concurrently while earlier ones are still being handled? Addressing these challenges is the focus of this work. Throughout, we assume that the cost function between locations satisfies the metric properties.

This connection to metric bipartite matching naturally extends beyond logistics. The 1-Wasserstein distance, a widely used tool for comparing probability measures in machine learning, can be expressed as a minimum-cost matching between empirical distributions (Villani, 2009; Peyré & Cuturi, 2019). It has found broad applications in generative modeling, domain adaptation, fairness, and distributional drift detection (Tolstikhin et al., 2018; Liu et al., 2018; Cao et al., 2019; Balaji et al.,

2020). Formally, for two probability measures $\mu$ and $\nu$ on a metric space $(\mathcal{X}, d)$, it is defined as

$$W_1(\mu, \nu) \; = \; \inf_{\pi \in \Pi(\mu, \nu)} \int_{\mathcal{X} \times \mathcal{X}} d(x, y) \, d\pi(x, y),$$

where $\Pi(\mu, \nu)$ denotes the set of couplings of $\mu$ and $\nu$.

In many practical scenarios, however, samples arrive in dynamic streams, so the optimal matching may change at every step. Exact recomputation quickly becomes infeasible in high-throughput applications such as real-time monitoring (Rabin et al., 2011), adaptive learning (Chen et al., 2018), or fairness auditing (Chouldechova, 2017). This challenge has spurred a growing body of work on extending Wasserstein distances to richer domains, including graphs, manifolds, and structured biological spaces, where data naturally resides beyond Euclidean geometry (Séjourné et al., 2021; Kolouri et al., 2021; Haasler & Frossard, 2024; Ju & Guan, 2025). Beyond machine learning, dynamic geometric matching also arises in diverse real-world applications, such as quantifying similarity between evolving datasets Alvarez-Melis & Fusi (2020), tracking longitudinal changes in patient data (e.g., MRI scans) Gramfort et al. (2015), and employing matching-based metrics, such as the Earth Mover's Distance, in time series analysis Cheng et al. (2021).

Despite its importance, research on minimum-cost matchings in dynamic settings remains limited and primarily focused on Euclidean spaces. For instance, Goranci et al. (2025) recently studied the *dynamic Euclidean bipartite matching* problem, where updates are allowed on both sides of the matching. Their algorithm achieves an $O(1/\delta)$-approximation with sublinear (in $n$) update time, and has applications such as monitoring distributional drift in streaming data. Also see Andoni et al. (2009) for a streaming algorithm with similar approximation guarantee under insertions and deletions. However, the framework of Goranci et al. (2025) assumes that both sides of the bipartite graph always contain the same number of vertices. Although the techniques could potentially be adapted to settings with one-sided arrivals or departures, such an extension would require substantial modifications to the proof, which currently relies on balanced instances. Moreover, the framework is designed for low-dimensional Euclidean spaces and does not naturally extend to higher-dimensional or more general metric spaces.

These limitations motivate the central question of this work:

> *Is it possible to design a fast, constant-factor approximate bipartite matching algorithm for insertions that works in any metric space?*

This question forms the core of our study. It motivates a formal treatment of incremental matchings in general metric spaces and establishes a bridge between practical applications and theoretical guarantees. We now formally define the problem.

**Problem 1 (Incremental Metric Bipartite Matching)** *Let $S$ be a fixed set of $n$ servers embedded in a metric space $(X, d)$. Requests $R = r_1, r_2, \ldots$ arrive online, one at a time. At time $t$, the algorithm has observed requests $r_1, \ldots, r_t$ and must maintain a matching $\mathbb{M}_t \subseteq S \times \{r_1, \ldots, r_t\}$ that pairs each request to a distinct server in $S$. The cost of a matching is defined as the sum of edge distances, and the objective is to maintain a matching whose cost is within a constant factor of the optimal at all times.*

## 1.1 Our Contributions

In this work, we resolve Problem 1 by presenting the **first constant-factor approximation algorithm for incremental metric bipartite matching** that achieves sublinear update time in the number of edges. To the best of our knowledge, this is the first algorithm that applies to *arbitrary* metric spaces while guaranteeing provably fast updates. Our main result is stated below.

**Theorem 1** *For any $0 < \delta \le 1$, there exists a deterministic algorithm that maintains an $O(1/\delta^\alpha)$-approximate solution for the incremental metric bipartite matching problem on sets $R$ and $S$ embedded in a metric space, with an update time of*

$$O\big(n^{1+\delta} \cdot \log^2\big(\tfrac{1}{\delta}\big) \cdot \log(n\Delta)\big),$$

*where $\alpha = \log_3 2$ and $\Delta$ is the aspect ratio of the metric space.*

The total execution time of our incremental algorithm matches the static algorithm of Agarwal & Sharathkumar (2014) while achieving the same approximation ratio. In this sense, our result **strictly generalizes** their work: it provides the same guarantees in the *static* setting while additionally supporting *dynamic insertions*.

In addition to its dynamic nature, our algorithm supports parallel request processing, allowing new requests to begin execution even while earlier ones are still running. This design avoids queue build-up and is well suited for many applications, since several core subroutines parallelize naturally. As a result, incoming requests can be handled concurrently rather than sequentially, making the algorithm particularly effective in batched-insertion scenarios. To complement the theoretical contribution, we provide efficient implementations on both CPU and GPU. In extensive empirical evaluations on synthetic and real-world datasets, we benchmark against standard baselines including a greedy algorithm and (for low-dimensional Euclidean spaces) quadtree-based greedy approaches. Across all settings, our implementations consistently match or outperform these baselines in running time while maintaining competitive solution quality.

## 1.2 RELATED WORK

Classical algorithms for bipartite matching scale poorly in the incremental setting. The Hungarian algorithm, built on a primal–dual framework Kuhn (1955); Munkres (1957), computes an exact minimum-cost matching in $O(n^3)$ time; even optimized variants under mild assumptions require $\tilde{O}(n^{2.5})$ time Duan & Pettie (2016). Maintaining optimality as new requests arrive is particularly expensive: each update requires a Hungarian search step costing $\Theta(n^2)$ in metric graphs, so processing $n$ requests sequentially leads to a total runtime of $O(n^3)$. Such bounds are prohibitive for large-scale systems. A recent breakthrough by Chen et al. (2022) achieves almost linear-time algorithms for minimum-cost flow in general graphs with polynomial weights. A related line of work shows that some of the techniques in the above paper have been adapted to obtain a $(1 - \varepsilon)$-approximate maximum-flow algorithm in the incremental edge-update setting with update time $m^{o(1)}\varepsilon^{-3}$ van den Brand et al. (2024). Our setting, however, is fundamentally different: we must maintain an approximate min-cost perfect matching under vertex arrivals under metric costs. Extending ideas that maintain approximate flows under incremental updates to instead maintain exact combinatorial structure (such as perfect matchings), while still achieving approximate cost guarantees, appears to be an extremely challenging open problem.

Approximation algorithms for minimum cost perfect matchings provide a way to circumvent these barriers. Agarwal & Sharathkumar (2014) gave a deterministic offline algorithm for metric spaces that constructs a $1/\delta^{0.631}$-approximate minimum-cost matching in $O(n^{2+\delta})$ time by combining distance scaling with simultaneous augmenting path searches. While this result shows that near-quadratic approximations are achievable in the offline metric setting, directly extending it to an incremental model is difficult: every new request may trigger searches over $\Theta(n^2)$ edges, and the distance scaling framework depends on a constant-factor estimate of the optimum, which is hard to maintain dynamically. More recently, in the context of metric optimal transport, advances have yielded $(1 + \delta)$-approximation algorithms running in near-quadratic time Zuzic (2023); Fox (2024).

In a different line of work, researchers have developed additive-approximate algorithms for optimal transport and related matching problems. These include entropic-regularization–based methods Cuturi (2013); Jambulapati et al. (2019) as well as combinatorial approaches Lahn et al. (2019; 2023) . A common feature of many of these methods is that they are highly parallelizable, and several admit efficient GPU implementations—for example, Sinkhorn-based approximations Cuturi (2013) and the push–relabel–based approximation of Lahn et al. (2023).

## 1.3 OVERVIEW OF TECHNIQUES

Our incremental algorithm builds on the static algorithm by Agarwal & Sharathkumar (2014). A key insight of the static algorithm is that it does not operate directly on the given metric $d(\cdot, \cdot)$. Suppose $\omega$ is a good 'guess' for the offline optimal solution and $0 < \delta \leq 1$ is a fixed parameter. We construct a hierarchy of $\mu = O(\log(1/\delta))$ progressively "scaled-down" metrics, $\mathbb{M}_0, \mathbb{M}_1, \ldots, \mathbb{M}_\mu$.

At the base level $\mathbb{M}_0$, each original distance $d(s, r)$ is rescaled by a factor of about $n/(\varepsilon\omega)$ $(\varepsilon = 1/\mu)$ and then rounded up to the nearest integer. In this way, a discretized metric is produced in which

all distances fall in a bounded integer range, and moreover the optimal matching becomes $O(n)$ in the scaled space. At higher levels ($i > 0$), the distances are repeatedly *shrunk* and *rounded* further, with the shrinkage factor being roughly $n^{3^i\delta}$. It can be observed that the shrinkage factor grows very quickly with $i$, essentially at an exponential rate.

Another key idea is to always maintain a 1-*feasible partial matching* at each level. Given a complete bipartite graph $G(R \cup S, R \times S)$ where each edge $(r, s) \in R \times S$ has a cost $c(r, s)$, the seminal work of Gabow & Tarjan (1989) introduced the idea of a 1-feasible matching, which can be used to find an approximate minimum-cost matching.

A matching $\mathbb{M}$ along with a set of dual weights $y(\cdot)$ on the vertices is called a *1-feasible matching* if the following two condition holds

$$y(r) + y(s) \le c(r, s) + 1, \quad (r, s) \notin \mathbb{M}, \tag{1}$$
$$y(r) + y(s) = c(r, s), \qquad (r, s) \in \mathbb{M}. \tag{2}$$

Given a 1-feasible matching $\mathbb{M}$, an edge $(r, s) \in R \times S$ is called an *admissible*, if either $(r, s)$ is in $\mathbb{M}$ or $y(r) + y(s) = c(r, s) + 1$.

At a high level, the algorithm works as follows. For any $i$ (initialized to $0$), it uses the algorithm by Gabow and Tarjan to compute a partial 1-feasible matching $\mathbb{M}^i$ and corresponding duals $y_i(\cdot)$ under the scaled metric $\mathbb{M}_i$. The algorithm essentially works in phases. Each phase uses a BFS-styled graph search procedure over the admissible subgraph to match several requests via *augmenting paths*. Duals are suitably adjusted when the search cannot proceed. A crucial twist introduced by Agarwal & Sharathkumar (2014) is that the procedure is halted at level $i$ once *every free vertex* in $R$ reaches a dual weight $y_i^{\max} = O(n^{3^i\delta})$.

All such free requests (along with free servers) are then promoted to level $i + 1$ and the algorithm now operates in the scaled metric space $\mathbb{M}_{i+1}$. A delicate analysis shows that the number of requests and servers promoted to any level $1 \le i \le \mu$ is upper bounded by $n^{1-\Omega(3^i\delta)}$. Hence, each graph search phase can take only $n^{2-\Omega(3^i\delta)}$ while the number of phases is upper bounded by $y_i^{\max}$ leading to the desired $O(n^{2+\delta})$ runtime for each level. In case there are free vertices promoted to level $\mu$, they are matched using the standard Hungarian algorithm. However, with a suitable choice of parameter, it can be shown that the number of such requests can be only about $n^{2/3}$ and hence the Hungarian algorithm can take at most $O(n^2)$ time.

**Our Incremental Algorithm.** To extend the above algorithm to the incremental setting, a natural approach is to explicitly maintain $O(\log(1/\delta))$ 'levels' of 1-feasible partial matchings. For instance, when a new request $r_j \in R$ arrives, we need to - (i) determine the level at which we should be matching it and (ii) efficiently modify the existing matchings and duals at various levels to reflect this change.

One natural strategy to handle both (i) and (ii) is to initialize the new request at level 0 and augment the partial matchings and duals at each level, akin to Agarwal & Sharathkumar (2014), and promoting requests to higher levels as their dual values reach $y_i^{\max}$ for any level $0 \le i \le \mu$. However, one serious caveat of this strategy is that we may end up searching through the entire graph just to process a single arriving request, which could lead to a prohibitive update time of $\Theta(n^2)$.

We overcome this major challenge by departing from an augmenting path-based approach to a *push-relabel* styled framework in order to maintain the partial matchings $\mathbb{M}^i$ at each level $0 \le i \le \mu$. Roughly speaking, a newly arrived request $r$ starts at level 0 and simply looks for an admissible edge in all levels of scaled metric spaces between 0 and $\mu$. We pick such an admissible edge $(r, s) \in R \times S$ arbitrarily and execute a 'push'. Specifically, we match $r$ to $s$, decrease the dual of the server by 1 to maintain 1-feasibility, and in case the server was already matched to some other request $r'$, we make $r'$ free. In case no admissible edge is found for $r$, we do a 'relabel' - we increase the dual of $r$ to an extent where one edge becomes admissible. However, if the dual of $r$ reaches $y_0^{\max}$, then $r$ is promoted to level 1 and the process continues. In general, this may free up a request at any level $i \le \mu$ in which case we simply continue the push-relabel from this level. One crucial invariant of our algorithm is that a server that is matched at any level $0 \le i \le \mu$ is only available to requests that are at level $i$ or higher.

The above description might suggest we scan $\Theta(n^2)$ edges even to push a single free request; while that can happen in the worst case, a careful amortized analysis yields $O(n^{1+\delta})$. For each level $i$ the total number of requests ever promoted to $i$ is $n_i \leq n^{1-\Omega(3^i\delta)}$ (by the scaled metric and our choice of $y_i^{\max}$). Each failed admissible-edge search for a free request at level $\geq i$ causes a request dual increment (a relabel) of at least 1, and server dual decrements can be charged to these request increments, so the total number of such searches is $O(y_i^{\max})$ before the request is either matched or promoted. Each search inspects $O(n_i)$ matched servers at levels $\geq i$ (plus one free server), and we assume a data structure that finds the nearest free server in $O(1)$ time (we never search servers matched at lower levels). Hence the total work for admissible-edge searches at level $i$ over all $n$ arrivals is $O(n_i^2 \cdot y_i^{\max}) = O(n^{2+\delta})$ for suitable constants. With only $O(\log(1/\delta))$ levels this yields the stated amortized update time.

## 2    PRELIMINARIES

We introduce notations and important definitions that would be required to describe our algorithm. Given a (possibly partial) matching $\mathbb{M} \subseteq R \times S$, we denote the total cost of all the edges in $\mathbb{M}$ by $w(\mathbb{M})$. Throughout this and the next subsection, we will assume that we are given a value $\omega$ satisfying $w(\mathcal{M}_j^\star) \leq \omega \leq 2w(\mathcal{M}_j^\star)$, where $\mathcal{M}_j^\star$ is the offline optimal solution of requests $r_1, r_2, \cdots r_j$ for any $1 \leq j \leq n$. We will show how to remove this assumption in Section 3.2.

$\omega$-**Scaled Metrics.** Given a parameter $\omega > 0$, $0 \leq \delta \leq 1$, we define $\mu + 1$ different finite metric spaces $\mathbb{M}_0, \ldots, \mathbb{M}_{\mu+1}$, where $\mu \leq \log_3(\frac{2-\delta}{9\delta})$ and each metric $\mathbb{M}_i$ is on the points $S \cup R$ equipped with a distance $\widehat{d}_i(\cdot, \cdot)$ defined as follows -

$$
\widehat{d}_i(s,r) = \begin{cases} \left\lceil \frac{2d(s,r)\cdot n}{\varepsilon\omega} \right\rceil & \text{if } i = 0, \\[4mm] \left\lceil \frac{\widehat{d}_{i-1}(s,r)}{2(1+\varepsilon)^2 n^{\varphi_{i-1}}} \right\rceil & \text{if } i > 0, \end{cases}
\tag{3}
$$

where $\varphi_i = 3^i\delta$, $\varepsilon = \frac{1}{2\log_3(1/\delta)}$,

Define $y_i^{\max} = \frac{30}{\varepsilon}n^{\varphi_i}, \forall i \in \{0, \ldots, \mu+1\}$

We state a two key properties of these metric hierarchy in the form of the following lemma.

**Lemma 1 ( Agarwal & Sharathkumar (2014))** *The following properties are true for the distance functions $\widehat{d}_i(\cdot, \cdot)$*

 1. *For $i \geq 0$, $\widehat{d}_i(\cdot, \cdot)$ is a metric.*

 2. *For $i \geq 1$, there is a scaling factor $\sigma_i$ such that $(1 - \varepsilon/3)\sigma_i\widehat{d}_i(s,r) \leq d(s,r) \leq \sigma_i\widehat{d}_i(s,r)$*

Our algorithm will have a notion of levels for each requests, servers, matchings between them and corresponding duals which corresponds to the hierarchical metric spaces defined above. For each level $i \in \{0, \ldots, (\mu+1)\}$, $\mathbb{M}^i$ will denote a partial matching at level $i$ and we define the following w.r.t $\mathbb{M}^i$. $B_S^i$ denotes set of servers that are matched in $\mathbb{M}^i$ and $\mathcal{B}_S^i = \bigcup_{j=i}^{\mu+1} B_S^j$, that is $\mathcal{B}_S^i$ is the set of servers matched in $\mathbb{M}^k$, $k \geq i$. $S^{\mathbb{F}}$ is the set of all free servers at any point in the algorithm. For any request or server in $R \cup S$, we will maintain an integer $level(\cdot)$ which will denote the level at which they are currently matched. Finally, let $y_i(\cdot)$ be dual weights on $R \cup S$ at level $i$.

We introduce two invariants maintained by our algorithm at all points of time.

 (I1) At each level $i$, the matching $\mathbb{M}$ maintained by the algorithm is 1-feasible. That is, at any level $i \in \{0, \cdots, \mu+1\}$

$$
\begin{aligned}
y_i(s) + y_i(r) &\leq \widehat{d}_i(s,r) + 1 && \text{where } (s,r) \notin \mathbb{M}^i \text{ and } s \in \mathcal{B}_S^i \\
y_i(s) + y_i(r) &= \widehat{d}_i(s,r) && \text{where } (s,r) \in \mathbb{M}^i
\end{aligned}
$$

**(I2)** For any unmatched server $s$ (that is $s \in S^{\mathbb{F}}$); $y_i(s) = 0, \forall i \in \{0, \cdots, (\mu + 1)\}$. For any request $r$, if $level(r) = i$, then $y_k(r) = y_k^{\max}, \forall 0 \leq k < i$. For any server $s$, if $level(s) = i$, then $y_k(s) = 0, \forall 0 \leq k < i$

## 3 INCREMENTAL PUSH-RELABEL ALGORITHM

In this section we give necessary details of our algorithm and a sketch of the analysis. The main pseudocode and detailed proofs can be found in Appendix A.

**Initialization.** At the start of the algorithm, all servers are placed in the free-server set $S^{\mathbb{F}}$, while the sets $B_S^i$ for levels $i \in \{0, \ldots, \mu+1\}$ are empty (i.e., $|B_S^i| = 0$ for all $i$). $level(s) = +\infty, \forall s \in S$ The initial matching is empty, denoted $\mathbb{M}_0 = \{\emptyset\}$. For all $i \in \{0, \ldots, (\mu + 1)\}$ and for all $s \in S^{\mathbb{F}}$, the dual $y_i(s) = 0$.

We now describe our incremental push–relabel algorithm for handling an arriving request $r_j$. The algorithm maintains the current matching $\mathbb{M}_{j-1}$, dual weights for requests and servers at each level, and data structures for admissible edges.

**Incremental-Push-Relabel.** Upon arrival of request $r_j$, its dual weights are initialized to zero across all levels, and it is marked as a free request $r^f$. We compute the $\omega$-scaled metric distances $\widehat{d}_i(r_j, \cdot)$ for all $0 \leq i \leq \mu + 1$ We create a sorted list $\mathcal{L}_{r^f}$ of free servers ordered by their distance $d(\cdot, r^f)$. Both $level(r^f)$ and counter $i$ are set to 0.

While there is a free request $r^f$ and $i \leq \mu + 1$:

- **Admissible edge search.** We query FIND-ADMISSIBLE-EDGE$(r^f, i)$.
  - If an admissible edge $(r^f, s)$ is found:
    * If $s$ is free, we insert $(r^f, s)$ into $\mathbb{M}_j$, and decrease $y_i(s)$ by 1. Moreover,
      · Extract $s$ from $\mathcal{L}_{r^f}$
      · Move $s$ from $S^{\mathbb{F}}$ to $B_S^i$
      · Set: $level(s) \leftarrow i$
    * If $s$ is already matched to some $r'$, we perform a *push*: replace edge $(r', s)$ with $(r^f, s)$ in $\mathbb{M}_j$, decrease $y_i(s)$ by 1, and perform the following:
      · Move $s$ from $B_S^{level(r')}$ to $B_S^i$
      · Set: $level(s) \leftarrow i, r^f \leftarrow r', i \leftarrow level(r')$
  - If no admissible edge is found, we perform a *relabel*: increase $y_i(r^f)$ by the minimum slack needed so that at least one edge becomes admissible. The minimum slack computation is as follows:
    * Let $s^f$ be the first server in $\mathcal{L}_{r^f}$
    * Then the minimum slack quantity is

$$\min_{s \in \mathcal{B}_S^i \cup \{s^f\}} \left\{ \widehat{d}_i(s, r) - y(r^f) - y(s) \right\} + 1.$$

- **Promotion.** If increment by minimum slack pushes $y_i(r^f)$ up to $y_i^{\max}$, then the request is promoted to upper level by setting : $level(r^f) \leftarrow i + 1, i \leftarrow i + 1$.

If there is free request promoted to level $\mu + 2$, match it to a server at level $\mu + 2$ or $S^{\mathbb{F}}$ using Hungarian Algorithm. The resulting matching is denoted $\mathbb{M}_j$.

**Find-Admissible-Edge.** Given request $r$ at level $i$, we first scan all servers currently matched at levels $\geq i$ (that is, servers $\in \mathcal{B}_S^i$) to check whether any edge $(r, s)$ is admissible at level $i$, If such a server exists, it is returned. Otherwise, we check the closest free server $s^f$ from sorted list $\mathcal{L}_r^i$ for admissibility under the same condition. If none exist, the procedure returns $\emptyset$.

We would like to re-emphasize that we significantly deviate from the algorithm and analysis of the static approximation algorithm of Agarwal & Sharathkumar (2014) in two major aspects. Firstly,

while the static algorithms build the levels successively by pushing both unmatched requests and free servers higher and higher, we need to maintain partial matchings at all the levels simultaneously. In fact, in our algorithm the level of a request increases while that of a server can only decrease. Secondly, as highlighted in Section 1.3, rather than using augmenting paths, we use a push-relabel framework to locally match and unmatch requests. Finding an admissible edge is the bottleneck in this operation. Crucially, we use the 1-feasibility property of the matchings and dual adjustments to pay for this expensive step.

## 3.1 ANALYSIS

**Cost.** The costs analysis of our algorithm follows that of the static algorithm by Agarwal & Sharathkumar (2014) while having a few crucial differences. The main distinction stems from the fact that in the static algorithm, while analyzing the cost of matching at any level $i$, the set of requests and servers promoted to level $i$ form a balanced bipartite subgraph of $R \cup S$. Due to the incremental nature of our setting, the graph we analyze at level $i$ is skewed - it may have more servers than requests promoted to level $i$. This introduces non-trivial modifications in the analysis. We provide all the details in Appendix A.2 and prove the following central lemma.

**Lemma 2** *Let $\mathbb{M}_j$ be the matching maintained by our algorithm after the insertion of $r_j$ and let $\mathcal{M}_j^\star$ be the offline optimal matching on that perfectly matches $r_1, r_2, \cdots r_j$ with servers in $S$. Then $w(\mathbb{M}_j) \leq O(1/\delta^\alpha) \cdot w(\mathcal{M}_j^\star)$.*

The cost bound relies on maintenance of the dual invariants (**I1**) and (**I2**) by our algorithm. While the static algorithm creates these duals in successive iterations, maintenance of these duals simultaneously for all levels is a novel contribution of this work.

**Update Time.** We sketch the update time analysis of our algorithm which forms the technical heart of our paper. Each update consists of three operations: *Relabel*, where the dual of the active request is increased; *Push*, where the algorithm either matches a free server or an already matched server from the same or higher level; and **Find-Admissible-Edge**, where the algorithm scans for a 1-admissible edge. We show that the total number of such operations over $n$ requests can be bounded by $O(n^{2+\delta} \log^2(1/\delta))$, which implies the desired amortized bound per arrival.

**Relabel operations.** At level $i$, each request $r$ enters with dual $y_i(r) = 0$ and increases monotonically until either it reaches the maximum allowed dual value $y_i^{\max}$ or is matched. Thus, the total number of relabel increments per request per level is upper bounded by $y_i^{\max}$.

**Push operations.** Push steps are more subtle because they may involve cascading reassignments of servers. To control this, we relate server dual decrements (triggered by pushes) to request dual increments (triggered by relabels). The following amortization allows us to bound the number of push operations by the number of relabel operations.

**Lemma 3** *For any level $i$, over all the insertions, the magnitude of server dual decrements is upper bounded by the magnitude of request dual increments.*

**Admissible-edge searches.** The most expensive operation naively is scanning for admissible servers, which could cost $\Theta(n)$ per request per level. However, we prove two crucial lemmas that will establish that the amortized number of operations is still bounded by $O(n^{2+\delta})$. The first key ingredient is the following lemma which upper bounds the number of requests that are promoted to level $i$ or higher.

**Lemma 4** *At any point in the insertion sequence, at level $i$, the number of requests (and hence the number of servers) matched at level $i$ or higher , denoted by $n_i$, is at most $n^{1-\Phi_i}$, where $\Phi_i = \sum_{k=0}^{i-1} \varphi_k = \frac{3^i-1}{2}\delta$.*

This bound intuitively implies that search time for admissible edge reduces drastically at higher levels. While a similar property holds for the static algorithm by Agarwal & Sharathkumar (2014), our search for an admissible edge also need to consider free servers which can be $\Theta(n)$ in the worst case. However, we overcome this by simply maintaining a list of servers for every request sorted according to distance. These two properties give us the following lemma.

**Lemma 5** *For a fixed level $i$, the total time spent in admissible-edge searches across all request insertions is $O(n^{2+\delta})$.*

***Proof.*** [Proof sketch] From the previous lemma, only $n^{1-\Phi_i}$ requests are matched at level $i$ or higher at any point of execution. For each such request, we bound the total number of search for an admissible edge by $2y_i^{\max}$ before the request is promoted to level $i+1$ (in case it is). To see this, recall the for a free request at level $i$, the algorithm scans through all the servers that are currently matched at level $i$ or higher plus the set of currently unmatched servers. The number of operations for one such search can be upper bounded using Lemma 4 by $n_i + 1$, where the additional operation happens for extracting the nearest unmatched server. A successful search can be charged to a push step, while an unsuccessful search is charged to a relabel step - both of which are upper bounded by $y_i^{\max}$ for any request at level $i$. This along with Lemma 4 proves the claim with suitable choice of parameters. □

The above analysis does not directly hold for level $\mu + 2$ since we are running Hungarian algorithm at that level. However, observe that by Lemma 4 $n_{\mu+2} < n^\delta$. Hence, each Hungarian search cannot take more than $n^{1+\delta}$ time. Details can be found in Appendix A.3.

**Lemma 6** *For a sequence of requests $r_1, r_2, \cdots r_j$, the total update time of our algorithm is upper bounded by $O(n^{2+\delta} \cdot \log^2(1/\delta))$.*

### 3.2 REMOVING THE ASSUMPTION ON $\omega$

Recall that throughout the previous section, we had assumed that we are given an estimate $\omega$ such that after the arrival of any request $r_j$, $w(\mathcal{M}_j^\star) \leq \omega \leq 2w(\mathcal{M}_j^\star)$, where $\mathcal{M}_j^\star$ is the offline optimal solution of requests $r_1, r_2, \cdots r_j$. In this section, we remove this assumption using a standard guess-and-double trick.

When the first request $r_1$ is inserted, the procedure begins with the initial value of $\omega = \min_{s \in S} d(s, r_1)$. Suppose that after processing $r_j$, we find that at some level $0 \leq i \leq (\mu + 1)$, the number of requests with level $i$ or higher is greater than $n^{1-\Phi_i}$ (where $\Phi_i = \frac{3^i - 1}{2}\delta$). Then we double the value of $\omega$ and compute an offline matching by artificially re-insert the request sequence $\{r_1, \ldots, r_j\}$ with the new value of $\omega$ using our algorithm. We are now ready to finish the proof of Theorem 1 as follows

Let us divide the insertion sequence $r_1, r_2, \cdots r_n$ in to $\ell$ phases $\mathcal{P}_0, \mathcal{P}_1, \mathcal{P}_\ell$ such that the value of $\omega = 1$ at the beginning of $\mathcal{P}_0$ and it was doubled at the beginning of $\mathcal{P}_k$ for any $k \geq 1$. For $0 \leq k \leq \ell$, let $\omega_k$ denote the value of $\omega$ at all time points in phase $\mathcal{P}_k$. We show in Appendix A.2 that the cost bound holds as long as $\omega$ is at most twice the value of the optimal solution. We claim that this property is always true for each phase. We prove this by induction on the number of phases $k$. Note that this property holds at the beginning of $\mathcal{P}_0$ by our choice of $\omega_0 = \min_{s \in S} d(s, r_1)$ and will also hold at the end of this phase since optimal is monotonic. Now fix any $\mathcal{P}_k, k > 1$ and let $r_j$ be the first request in this phase. By the condition of doubling, $\omega_{k-1} < w(\mathcal{M}_j^\star)$ and hence $\omega_k = 2\omega_{k-1} < 2w(\mathcal{M}_j^\star)$. The property holds for all request insertions in this phase by monotonicity of optimal matching.

For the runtime bound, note that our algorithm always doubles $\omega$ when we find that at some level $0 \leq i \leq (\mu + 1)$, the number of requests with level $i$ or higher is greater than $n^{1-\Phi_i}$. We show in the Appendix A.3 that if this is true, the amortized runtime for a phase is upper bounded by $O(n^{2+\delta} \log^2(1/\delta))$. The only thing remaining to show is that the number of phases is upper bounded by $\log(n\Delta)$. This follows from the fact that $w(\mathcal{M}_n^\star) \leq n\Delta$ and hence we are done.

## 4 EXPERIMENTS

In this section we present our experimental results. We developed two independent implementations of our algorithm[1]. The first one is implemented using C++ and performs all operations on the CPU. The second leverages PyTorch, offloading the computationally intensive components of the

---

[1]Our code is available on https://github.com/ritesh-777/Incremental-Metric-Bipartite-Matching-Algorithm.

algorithm to a GPU. All the tests are performed on a machine with AMD EPYC 7763 64-Core Processor and 514 GB of RAM using a single computational thread for CPU bounded tasks. For the GPU bounded tasks we have used NVIDIA A100-SXM4 Graphics processor with 40GB GPU memory belonging to the same machine.

**Datasets:** We evaluate our algorithm on three real-world datasets and one synthetic dataset, each consisting of 10,000 data points. **(i) MNIST.** The MNIST dataset ( LeCun et al.) contains about 70,000 handwritten digit images, each represented as a $28 \times 28$ grayscale grid (784-dimensional). We sample two distributions, normalize each image so that pixel intensities sum to one, and measure distances using the $L_1$ norm. **(ii) NYC-Taxi.** The New York City Taxi dataset H (2021) provides pickup and drop-off locations. We construct two distributions from trips completed during the first week of a given month, ordering requests by `pickup_datetime` to capture sequential arrivals. **(iii) Beijing Road Network.** The Beijing Road Network, extracted from large-scale taxi GPS trajectory data in Lian & Zhang (2018) and subsequently processed in Jain et al. (2021), comprises approximately 31,000 nodes and 72,000 edges. In our experiments, we employ the shortest-path metric as the underlying distance function. **(iv) Synthetic.** We also generate 10,000 points uniformly at random in the two-dimensional domain $[0, 100]^2$. Additional results for this dataset appear in Appendix B. For NYC-Taxi data and Synthetic data, we employ Euclidean distances.

**Adaptation to batch (Batch Incremental PR):** Although Algorithm 3 is inherently sequential, both of the crucial parts - push and relabel - lend themselves to parallelization. More specifically, we create an admissible graph with all free requests in parallel. Further, the push step can be executed by adapting a parallel implementation for computing a maximal matching on the admissible graph. Further, the relabel step involves updating entries in a certain 'slack matrix' which can also be executed in parallel. We adapt the parallel implementation of push-relabel from Lahn et al. (2023) who give an $\varepsilon$-additive approximation for optimal transport (and hence also min-cost perfect matching) in the static setting. For our experiments, we assume that requests are arriving in batches of 200 and all requests of a specific batch are processed using the parallel implementation on a GPU. We provide additional experiments in Appendix B.2 showing how the average update time changes with varying batch sizes.

**Tests:** For comparison, we evaluate our Batch Incremental PR algorithm against Greedy, QuadTree-based (QT) and Dynamic Euclidean (Goranci et al. (2025)) baselines. Each dataset is sampled 10 times, and the server size is fixed at 10,000. For varying request sizes $n \in \{1000, 2000, \ldots, 10000\}$, we report the average matching cost and average amortized runtime of all algorithms. In our experiments, we set $\delta = 0.001$. Furthermore, in Section B.1, we provide a detailed sensitivity analysis of $\delta$, highlighting its role in governing the runtime–cost trade-off. In Greedy, for any newly arrived request, the algorithm chooses the nearest free server. In QT (Har-Peled (2011)), we build a (randomly-shifted or deterministically-shifted) quadtree over the point sets, process tree nodes bottom-up: at each cell, greedily match as many red/blue points inside the same cell as possible and unmatched points are propagated (pushed) up to parent cells and matched there (again greedily). We have used CPU based implementation for QT. For Greedy, we have used GPU to compute distance which gives benefits to Greedy process high dimensional data points. We used the open-source implementation of Dynamic Euclidean provided by Zheng (2024) on the NYC-Taxi and Synthetic datasets, as both lie in Euclidean space. For all experiments, we set the branching factor to 16. However, for Beijing Road Network, we precomputed pairwise shortest path distance since shortest-path computation adds a massive computational overhead.

**Results:** In terms of runtime, Batch Incremental PR consistently outperforms Greedy on both MNIST and NYC-Taxi datasets (Figure 1(b), Figure 2(b)). In contrast, both QT and Dynamic Euclidean achieve faster runtime on the Taxi dataset (Figure 2(b)), albeit at the expense of higher matching cost. With respect to cost, Batch Incremental PR consistently outperforms both QT and Dynamic Euclidean on the Taxi dataset (Figure 2(a)), while performing comparably to Greedy. On the MNIST dataset, the algorithms QT and Dynamic Euclidean are inapplicable since distances are computed using $L_1$-norm. Furthermore, Batch Incremental PR surpasses Greedy with a significant margin both in terms of cost and update time. For the Beijing Road Network, QT and Dynamic Euclidean are not applicable because the metric is based on shortest-path distances. Batch Incremental PR consistently achieves a substantially lower cost and maintains a significant margin over Greedy (Figure 3(a)). In terms of runtime, Batch Incremental PR outperforms greedy till about 7,000 data points (Figure 3(b)) after which Greedy becomes better. However, the average response time of Batch Incremental PR never exceeds 50 milliseconds. Overall, Batch Incremental PR consistently

achieves the best balance between cost and runtime across datasets, and crucially, its performance advantage extends beyond low-dimensional Euclidean settings.

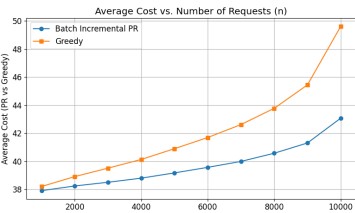 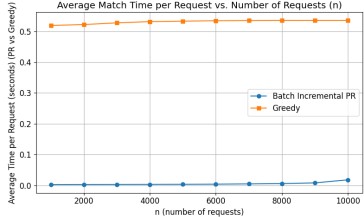

(a) Average cost per request      (b) Average time per request (in seconds)

Figure 1: Plots of MNIST data

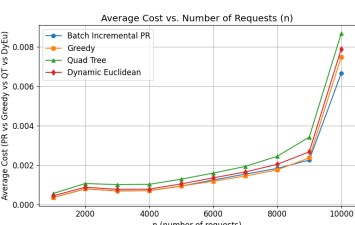 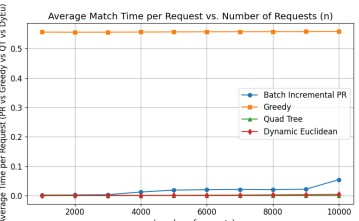

(a) Average cost per request      (b) Average time per request (in seconds)

Figure 2: Plots of NYC-Taxi data

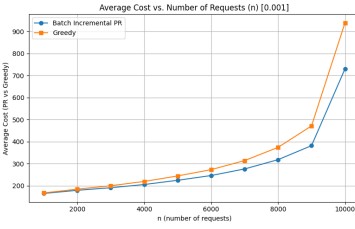 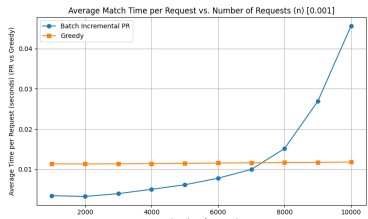

(a) Average cost per request      (b) Average time per request (in seconds)

Figure 3: Plots of Beijing Road Network

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

## APPENDIX

## A    DETAILS OF ALGORITHM AND ANALYSIS

### A.1    ANALYSIS

Our analysis consists mainly of two parts. In section A.2, we establish the competitive ratio of algorithm 1. Next, in section A.3, we show that the total runtime of the Algorithm 1 is $\widetilde{O}(n^{2+\delta})$. We

---

**Algorithm 1** INCREMENTAL-PUSH-RELABEL

---

**Input:** The $j$-th request $r_j$, where $j \in \{1, \ldots, n\}$
**Output:** Matching $\mathbb{M}_j$ after macthing $r_j$
1: $\mathbb{M}_j \leftarrow \mathbb{M}_{j-1}$
2: $y_i(r_j) \leftarrow 0$ for all $i \in \{0, \ldots, \mu\}$
3: $r^f \leftarrow r_j$        $\triangleright$ free request
4: $\mathcal{L}_{r^f} \leftarrow$ CREATE-SORTED-EDGE-LIST$(r^f)$
5: $i \leftarrow 0$
6: **while** $r^f \neq \emptyset$ and $i \leq (\mu + 1)$ **do**
7:      **if** $y_i(r^f) = y_i^{\max}$ **then**
8:          $i \leftarrow i + 1$        $\triangleright$ promotion of $r^f$
9:          $level(r^f) \leftarrow i$
10:          **continue**
11:      **end if**
12:      $s \leftarrow$ FIND-ADMISSIBLE-EDGE$(r^f, i)$
13:      **if** $s \neq \emptyset$ **then**
14:          **if** $s \in S^{\mathbb{F}}$ **then**        $\triangleright$ free server
15:              Add edge $(s, r^f)$ to $\mathbb{M}_j$
16:              $y_i(s) \leftarrow y_i(s) - 1$        $\triangleright$ Relabel : decrease server dual
17:              $r^f \leftarrow \emptyset$
18:              $level(s) \leftarrow i$
19:              $S^{\mathbb{F}} \leftarrow S^{\mathbb{F}} \setminus \{s\}$
20:              Remove the first element from $\mathcal{L}_{r^f}$        $\triangleright$ update $\mathcal{L}_{r^f}$
21:              $B_S^i \leftarrow B_S^i \cup \{s\}$
22:          **else**
23:              Add $(s, r^f)$ to $\mathbb{M}_j$        $\triangleright$ Push : server $s$ matched to $r'$
24:              let $r'$ be the request matched to $s$
25:              Remove $(s, r')$ from $\mathbb{M}_j$
26:              adjust the server dual: $y_i(s) \leftarrow y_i(s) - 1$
27:              $r^f \leftarrow r'$
28:              $level(s) \leftarrow i$
29:              $B_S^{level(r')} \leftarrow B_S^{level(r')} \setminus \{s\}$
30:              $B_S^i \leftarrow B_S^i \cup \{s\}$
31:              $i \leftarrow level(r')$
32:          **end if**
33:      **else**
34:          Let $s^f$ be the first server in $\mathcal{L}_{r^f}^i$
35:          $slack_{\min} \leftarrow \min_{s \in \mathcal{B}_S^i \cup \{s^f\}} \left\{ \widehat{d}_i(s, r) - y(r^f) - y(s) \right\}$
36:          $y_i(r^f) \leftarrow \min\{(y_i(r^f) + slack_{\min} + 1), y_i^{\max}\}$    $\triangleright$ Relabel: increase request dual
37:      **end if**
38: **end while**
39: **if** $r^f \neq \emptyset$ and $level(r^f) = (\mu + 2)$ **then**
40:      Use Hungarian algorithm to match $r^f$ with servers in the set $\mathcal{B}_S^{\mu+2} \cup S^{\mathbb{F}}$
41: **end if**
42: **return** $\mathbb{M}_j$

---

**Algorithm 2** CREATE-SORTED-EDGE-LIST$(r)$

---

**Input:** Request $r$
**Output:** Sorted list of free servers $\in S^{\mathbb{F}}$ over $d(\cdot, r)$
1: $\mathcal{L} \leftarrow \{\emptyset\}$
2: **for** $s \in S^{\mathbb{F}}$ **do**
3:      $\mathcal{L} \leftarrow \mathcal{L} \cup \{(s, d(s, r))\}$
4: **end for**
5: SORT$(\mathcal{L})$ based on $d(\cdot, r)$
6: **return** $\mathcal{L}$

---

---

**Algorithm 3** FIND-ADMISSIBLE-EDGE$(r, i)$

---

**Input:** Request $r$ and its current level $i$
**Output:** A server $s$ having admissible edge with $r$ if exists
 1: **for** $s \in \mathcal{B}_S^i$ **do**                                 ▷ Find an admissible non-free server
 2:      **if** $y_i(s) + y_i(r) = \widehat{d}_i(s, r) + 1$ **then**
 3:          **return** $s$
 4:      **end if**
 5: **end for**
 6: Let $s^f$ be the first server in $\mathcal{L}_r$                       ▷ Find an admissible free server
 7: **if** $y_i(s^f) + y_i(r) = \widehat{d}_i(s^f, r) + 1$ **then**
 8:      **return** $s^f$
 9: **end if**
10: **return** $\emptyset$

---

state two easy but crucial observations about the algorithm INCREMENTAL-PUSH-RELABEL which will be used at various parts in the analysis.

**Observation 1** *For any $1 \le j \le n$, the algorithm maintains a matching $\mathbb{M}_j$ of the request sequence $\{r_1, r_2, \cdots r_j\}$. Moreover, once a server is matched, it never becomes free.*

The first observation follows from the simple fact that we run the main loop until there exists a free request (recall that requests can become free and matched during the processing of a new request). It is unclear at this point why the process should terminate. However, we prove in Section A.3 that it indeed does.

Given a matching $\mathbb{M}_j$, define $\mathbb{M}_j^i$ as the set of edges $(s, r) \in \mathbb{M}_j$ such that $level(r) = level(s) = i$ - we refer to such an edge as a *matched edge at level $i$*.

**Observation 2** *For any $r \in R$, $level(r)$ is monotonically increasing while for any $s \in S$, $level(s)$ is monotonically decreasing over the sequence of insertions.*

Both the observations follow from the fact that in FIND-ADMISSIBLE-EDGE , for any request $r$ the algorithm only considers edges to $s$ that $level(r) \le level(s)$.

We prove the following invariants for any matching $\mathbb{M}_j^i, 1 \le j \le n$, introduced in Section 2.

> **(I1)** At each level $i$, the matching $\mathbb{M}_j^i$ maintained by the algorithm is 1-feasible. That is, at any level $i \in \{0, \cdots, (\mu + 1)\}$
> $$y_i(s) + y_i(r) \le \widehat{d}_i(s, r) + 1 \quad \text{where } (s, r) \notin \mathbb{M}_j^i \text{ and } s \in \mathcal{B}_S^i$$
> $$y_i(s) + y_i(r) = \widehat{d}_i(s, r) \qquad \text{where } (s, r) \in \mathbb{M}_j^i$$

> **(I2)** For any unmatched server $s$ (that is $s \in S^{\mathbb{F}}$); $y_i(s) = 0, \forall i \in \{0, \cdots, (\mu + 1)\}$. For any request $r$, if $level(r) = i$, then $y_k(r) = y_k^{\max}, \forall 0 \le k < i$. For any server $s$, if $level(s) = i$, then $y_k(s) = 0, \forall 0 \le k < i$

**Lemma 7 ( Invariant (I2))** *For any unmatched server $s \in S^{\mathbb{F}}$, $y_i(s) = 0, \forall 0 \le i \le (\mu + 1)$. For any request $r$, if $level(r) = i$, then $y_k(r) = y_k^{\max}, \forall 0 \le k < i$. For any server $s$, if $level(s) = i$, then $y_k(s) = 0, \forall 0 \le k < i$.*

***Proof.*** The first claim follows from Observation 1 and the fact that for a server in $S^{\mathbb{F}}$, $y_i(s)$ is set to 0 in the initialization phase.

The second claim can be proved using induction on $i$. For the base case ($i = 0$), the lemma holds vacuously. Now consider any level $i \ge 1$. Firstly, by induction hypothesis, $y_k(r) = y_k^{\max}, 0 \le k < i - 1$. Further, since $r$ is matched at level $i$, there exists some iteration when it became a free vertex at level $i$. This implies $y_{i-1}(r)$ was set to $y_{i-1}^{\max}$ at some iteration. Furthermore, dual increments of $r$ happen only at the relabel step and the only dual values that are modified in subsequent iterations are $y_k(r)$ for $k \ge i$. The third claim follows from an analogous argument for the servers. $\qquad\square$

**Lemma 8 (Invariant (I1))** *The matching $\mathbb{M}$ maintained by Algorithm 1 is always 1-feasible.*

**Proof.** We proceed by induction over updates to $\mathbb{M}$. Suppose 1-feasibility holds after processing $r_{j-1}$. Consider any dual update during the processing of $r_j$. Duals change only in three places: two for servers and one for requests. We argue these two cases using contradiction.

*Case 1 (request update).* For contradiction, let us assume Invariant (**I1**) is violated for request $r$ at a point of time when $level(r) = i$. Recall that the dual variable $y_i(r)$ is increased by at least 1 if and only if there is no admissible edge from any $s \in \mathcal{B}_S^i \cup \mathcal{L}_r$. We want to show, $r$ maintains feasibility condition with all the servers of $\mathcal{B}_S^i \cup \mathcal{L}_r$. We can ignore all server $s' \in \mathcal{B}_S^k$ where $k < i$, since Observation 2 eliminates the possibility of $(s', r)$ being a matching edge in future. If $r$ is a matched request, its dual does not increase. Thus, $r$ is a free request at level $i$ with no admissible edge. After increasing $y_i(r)$, suppose for contradiction that feasibility is violated on some $(s, r)$, where $s \in \mathcal{B}_S^i \cup \mathcal{L}_r$, i.e.,

$$y_i(s) + y_i(r) \geq d_i(s, r) + 2.$$

By induction hypothesis, just before the update we had equality $y_i(s) + y_i(r) = d_i(s, r) + 1$, so $(s, r)$ was admissible. Since $r$ was free, the algorithm would have matched it, contradiction.

*Case 2 (server update).* When a request $r$ with $level(r) = i$ matches a server $s \in \mathcal{B}_S^i$, the dual of $s$ is decreased by 1. By induction hypothesis $(s, r)$ was non-matching 1-feasible edge before the update. Hence, decreasing $y_i(s)$ cannot violate feasibility.

Thus all updates preserve 1-feasibility. $\square$

## A.2 COST ANALYSIS

After matching $r_j$, let $\mathbb{M}_j$ and $\mathcal{M}_j^\star$ be the matching by the algorithm and the optimal matching (under $d(\cdot, \cdot)$), respectively. $w(\mathbb{M}_j)$ and $w(\mathcal{M}_j^\star)$ denote the cost of the respective matching under metric $d(\cdot, \cdot)$. Similarly, $\widehat{w}(\mathbb{M}_j)$ and $\widehat{w}(\mathcal{M}_j^\star)$ denote the cost of the respective matching under metric $\widehat{d}(\cdot, \cdot)$. We shall show that $w(\mathbb{M}_j) = O(1/\delta^\alpha)w(\mathcal{M}_j^\star)$. For our analysis, we assume that we maintain the guess $w$ such that $w(\mathcal{M}_j^\star) \leq \omega \leq 2w(\mathcal{M}_j^\star); \forall j \in \{1, \ldots, n\}$.

First, let us explore a few properties of the metric space defined by the distance function $\widehat{d}_i(\cdot, \cdot)$.

Let $\mathbb{M}_j^i \subseteq \mathbb{M}_j$ denotes the set of matching edges at level $i$. Let $S_j^i = \mathcal{B}_S^i \cup S^{\mathbb{F}}$. Let $R_j^i$ be requests such that for any $r \in R_j^i$; $level(r) \geq i$.

Lemma 9, 10, 11 follows from Agarwal & Sharathkumar (2014). However their notion of server sets used in the lemma are different ours. Specifically, $S_j^i$ contains servers which are both free or matched at any level $i$ or higher. Although the methodology of our proof is more or less similar to them, our proof departs at certain places from them due to different definition of server set. For the sake of completeness and readability, we present the full proof of the said Lemmas.

**Lemma 9** *1. For any $i \geq 0$, $d_i(\cdot, \cdot)$ is a metric.*

*2. For $i \geq 1$, $d_i(s, r) \geq \frac{6}{\varepsilon}$ for any $(s, r) \in S_j^i \times R_j^i$.*

*3. For any $i \geq 1$, there is a scaling factor $\sigma_i$ such that*

$$(1 - \varepsilon/3)\sigma_i d_i(s, r) \leq d(s, r) \leq \sigma_i d_i(s, r)$$

**Proof.** *Proof of part (i)* Given three points $a, b$ and $c$ in the metric space defined by the distance function $d(\cdot, \cdot)$, from the triangle inequality

$$d(a, b) + d(b, c) \geq d(a, c)$$

The inequality holds even if we multiply it by some $k \in \mathbb{R}$ as follows

$$\lceil kd(a, b) \rceil + \lceil kd(b, c) \rceil \geq \lceil kd(a, c) \rceil$$

Using this scaling property of the metric space, we will complete the proof of this part. The distance function is defined by $\widehat{d}_i(\cdot, \cdot)$. We use induction on $i$. When $i = 0$, we set $k = \frac{n}{\varepsilon\omega}$, where $n$ is

the total number of requests. We have $\widehat{d}_0(\cdot,\cdot)$ satisfying the triangle inequality. Assume $\widehat{d}_{i-1}(\cdot,\cdot)$ satisfies the triangle inequality. Setting $k = \frac{1}{2(1+\varepsilon)^2 n^{\varphi_{i-1}}}$ and $d(\cdot,\cdot) = \widehat{d}_{i-1}(\cdot,\cdot)$, we have $\widehat{d}_i(\cdot,\cdot)$ satisfying the triangle inequality.

*Proof of part (ii)* Consider request $r \in R_j^i$ is matched at level $i$. Then for any level $0 \le k < i$, $y_k(r) = y_k^{\max}$ following Invariant (**I2**). Moreover Invariant (**I2**) ensures, for any server $s \in S_j^i$, $y_k(s) = 0$. Then, for any edge $(s,r) \in S_j^i \times R_j^i$, the following holds,

$$y_k(s) + y_k(r) = \frac{30}{\varepsilon}\, n^{\varphi_k} \le \widehat{d}_k(s,r) + 1$$

This implies, $\widehat{d}_k(s,r) \ge \frac{30}{\varepsilon}\, n^{\varphi_k} - 1$. So, $\forall i > 0$, we have

$$\widehat{d}_i(s,r) \ge \frac{\widehat{d}_{i-1}(s,r)}{2(1+\varepsilon)^2 n^{\varphi_{i-1}}} \ge \frac{3}{2(1+\varepsilon)^2 \varepsilon} - 1 \ge \frac{6}{\varepsilon}$$

since $\varepsilon = \frac{1}{2\log_3\left(\frac{1}{\delta}\right)} \le \frac{1}{2}$

*Proof of part (iii)* Ignoring the ceiling operator in the scaling of distance only decrease the value of $\widehat{d}_i(\cdot,\cdot)$, so using 3 repeatedly, we obtain

$$\frac{1}{\sigma_i} \cdot d(s,r) \le \widehat{d}_i(s,r)$$

where, $\sigma_i = \frac{\omega\varepsilon(2(1+\varepsilon)^2)^i}{n^{1-\Phi_i}}$ . On the other hand,

$$\widehat{d}_i(s,r) \le \frac{1}{2(1+\varepsilon)^2 n^{\varphi_{i-1}}}\widehat{d}_{i-1}(s,r) + 1$$

By expanding the recurrence and performing the necessary algebraic manipulations, we arrive at

$$\widehat{d}_i(s,r) \le \frac{d(s,r)}{\sigma_i} + 2$$

Now using part (ii) of the lemma, we obtain

$$(1 - \varepsilon/3)\sigma_i\widehat{d}_i(s,r) \le d(s,r) \le \sigma_i\widehat{d}_i(s,r).$$

$\square$

**Corollary 1** *For $i \ge 1$, let $M$ and $M'$ be two (possibly partial) matchings of $S_j^i$, $R_j^i$.*

1. *If $M$ is a perfect matching, then $\widehat{w}(M) \ge \frac{6}{\varepsilon}\gamma_i$, where $\gamma_i$ is the number of requests in $R_j^i$.*

2. *$\left(1 - \frac{\varepsilon}{3}\right) \frac{\widehat{w}(M)}{\widehat{w}(M')} \le \frac{w(M)}{w(M')} \le \frac{1}{1-\frac{\varepsilon}{3}}\, \frac{\widehat{w}(M)}{\widehat{w}(M')}$*

For the next lemma, let $\mathcal{M}_j^i$ denotes the optimal matching between $S_j^i$ and $R_j^i$ under metric $d(\cdot,\cdot)$. This implies $\mathcal{M}_j^0 = \mathcal{M}_j^\star$.

**Lemma 10** *For all $i \in \{0, \cdots, (\mu+1)\}$, the following holds*
$$w(\mathbb{M}_j^i) \le (1 + 2\varepsilon)w(\mathcal{M}_j^i)$$

**Proof.** Observe that, for every edge $(s,r) \in \mathbb{M}_j^i$ ; $y_i(s) + y_i(r) = \widehat{d}_i(s,r)$ and for every vertex $v \in S_j^{i+1} \cup R_j^{i+1}$, $y_i(v) \ge 0$ (follows from Invariant (**I2**)). Therefore,

$$\widehat{w}(\mathbb{M}_j^i) \le \sum_{v \in S_j^i \cup R_j^i} y_i(v)$$

For every edge $(s,r) \in \mathcal{M}_j^i$ , $y_i(s) + y_i(r) \le \widehat{d}_i(s,r) + 1$. Every vertex of $R_j^i$ is incident on exactly one edge of $\mathcal{M}_j^i$ , so

$$\sum_{v \in S_j^i \cup R_j^i} y_i(v) \le \sum_{(s,r) \in \mathcal{M}_j^i} \widehat{d}_i(s,r) + \gamma_i \le \widehat{w}(\mathcal{M}_j^i) + \gamma_i \tag{4}$$

Consequently,

$$\widehat{w}(\mathbb{M}_j^i) \le \widehat{w}(\mathcal{M}_j^i) + \gamma_i \tag{5}$$

We prove the lemma for cases $i = 0$ and $i > 0$ separately:

**Case $i = 0$:** From the equation 5, we have $\widehat{w}(\mathbb{M}_j^0) \le \widehat{w}(\mathcal{M}_j^0) + j$. Since $\widehat{d}_0(s,r) = \left\lceil \frac{2d(s,r) \cdot n}{\varepsilon \omega} \right\rceil$ and $\omega \le 2(\mathcal{M}_j^0)$, we obtain the following inequalities:

$$\begin{aligned}
\widehat{w}(\mathbb{M}_j^0) &\le \sum_{(s,r) \in \mathcal{M}_j^0} \left( \frac{2n}{\varepsilon \omega} d(s,r) + 1 \right) + j \\
&\le \frac{2n}{\varepsilon \omega} w(\mathcal{M}_j^0) + 2j.
\end{aligned} \tag{6}$$

$$\widehat{w}(\mathbb{M}_j^0) \ge \frac{2n}{\varepsilon \omega} \sum_{(s,r) \in \mathbb{M}_j^0} d(s,r) = \frac{2n}{\varepsilon \omega} w(\mathbb{M}_j^0) \tag{7}$$

Combining the above equation 6 and 7 and using $\omega \le 2w(\mathcal{M}_j^0)$,

$$w(\mathbb{M}_j^0) \le w(\mathcal{M}_j^0) + \varepsilon \omega \le (1 + 2\varepsilon) w(\mathcal{M}_j^0).$$

**Case $i > 0$:** From 5 and Corollary 1(1) , we get

$$\widehat{w}(\mathbb{M}_j^i) \le \widehat{w}(\mathcal{M}_j^i) + \gamma_i \le (1 + \varepsilon/6) \widehat{w}(\mathcal{M}_j^i).$$

Finally by Corollary 1(2),

$$w(\mathbb{M}_j^i) \le \frac{1 + \varepsilon/6}{1 - \varepsilon/3} w(\mathcal{M}_j^i) \le (1 + \varepsilon) w(\mathcal{M}_j^i).$$

$\square$

**Lemma 11** *For all $i \in \{0, \cdots, (\mu + 1)\}$, the following holds*
$$w(\mathcal{M}_j^{i+1}) \le 3(1 + \varepsilon) w(\mathcal{M}_j^i)$$

**Proof.** Observe that $\mathbb{M}_j^i \oplus \mathcal{M}_j^i$ results in a set of vertex disjoint alternating cycles and alternating paths. For our purpose, we only care about the set of paths denoted by $\mathbb{P}$. Each path in $\mathbb{P}$ connects a some server of $S_j^{i+1}$ to a request in $R_j^{i+1}$ i.e. $|\mathbb{P}| = \gamma_{i+1}$. Using lemma 10,

$$\sum_{P \in \mathbb{P}} \sum_{(s,r) \in P} d(s,r) = w(\mathbb{M}_j^i) + w(\mathcal{M}_j^i) \le (2 + 2\varepsilon) w(\mathcal{M}_j^i) < 2(1 + \varepsilon) w(\mathcal{M}_j^i)$$

Recall that $d(\cdot, \cdot)$ satisfies the triangle inequality, so if the endpoints of a path $P_k \in \mathbb{P}$ are $(s_k, r_k) \in S_j^{i+1} \times R_j^{i+1}$, then $d(s_k, r_k) \le \sum_{(s,r) \in P_k} d(s,r)$. Hence,

$$w(\mathcal{M}_j^{i+1}) \le \sum_{P \in \mathbb{P}} \sum_{(s,r) \in P} d(s,r) \le 2(1 + \varepsilon) w(\mathcal{M}_j^i)$$

$\square$

**Theorem 2** *Let $\mathbb{M}_n$ be the final matching and $\mathcal{M}_n^\star$ be the optimal matching, then*

$$w(\mathbb{M}_n) = O(1/\delta^\alpha)w(\mathcal{M}_n^\star)$$

*, where $\alpha = \log_3 2 \approx 0.631$*

**Proof.** Using lemma 10,

$$w(\mathbb{M}_n) = \sum_{i=0}^{\mu+1} w(\mathbb{M}_n^i) \le (1+2\varepsilon) \sum_{i=0}^{\mu+1} w(\mathcal{M}_n^i) < 2(1+\varepsilon) \sum_{i=0}^{\mu+1} w(\mathcal{M}_n^i)$$

By applying lemma 11 repeatedly, we obtain,

$$w(\mathcal{M}_n^i) \le 2^i(1+\varepsilon)^i \sum_{i=0}^{\mu+1} w(\mathcal{M}_n^0) = 2^i(1+\varepsilon)^i \sum_{i=0}^{\mu+1} w(\mathcal{M}_n^\star)$$

Hence,

$$w(\mathbb{M}_n) < 2(1+\varepsilon)w(\mathcal{M}_n^\star) \sum_{i=0}^{\mu+1} 2^i(1+\varepsilon)^i \le (2(1+\varepsilon))^{\mu+2}w(\mathcal{M}_n^\star)$$

Putting the values of $\mu$ and $\varepsilon$, we get $(1+\varepsilon)^{\mu+2} = O(1)$ and $2^{\mu+2} = O(1/\delta^\alpha)$. Hence $w(\mathbb{M}_n) = O(1/\delta^\alpha)w(\mathcal{M}_n^\star)$ □

### A.3 UPDATE TIME ANALYSIS

Now, let us focus on the run-time analysis. Recall the notations one more time. After $n$ requests has arrived, $\mathbb{M}_n$ denotes the online solution and $\mathcal{M}_n^\star$ is the offline optimal solution. $\mathbb{M}_j^i \subseteq \mathbb{M}_j$ denotes the set of matching edges at level $i$. $S_j^i = \mathcal{B}_S^i \cup S^{\mathbb{F}}$ and $R_j^i$ is the set requests such that for any $r \in R_j^i$; $level(r) \ge i$. $\mathcal{M}_n^i$ denotes the optimal matching between $S_n^i$ and $R_n^i$ under metric $d(\cdot,\cdot)$. This implies $\mathcal{M}_n^0 = \mathcal{M}_n^\star$.

We compute the total running time of INCREMENTAL-PUSH-RELABEL over arrival of all requests. Recall that we are assuming that our metric space has bounded aspect ratio $\Delta$. This impacts the cost by a factor of $\log(n\Delta)$.

To process $j$-th request, INCREMENTAL-PUSH-RELABEL performs four major operations regardless of levels :

- **Creating sorted edge list :** For each newly arrived request $r$, the algorithm constructs the lists $\mathcal{L}_r$; using CREATE-SORTED-EDGE-LIST .
- **Push operation.** This step involves modifying the current matching by either adding or removing edges. Each addition or removal of an edge is accompanied by a corresponding decrease in the dual variable associated with the server of that edge.
- **Relabel operation.** In this step, the dual variable of a request is incremented.
- **Finding admissible edges.** For a free request $r$, FIND-ADMISSIBLE-EDGE either identifies a server from the set $\mathcal{B}_S^i \cup \mathcal{L}_r$ that forms an admissible edge with $r$, or reports that no such server exists.

Now we proceed with the analysis the following way. In lemma (Lemma 15) we bound the total runtime of the Relabel operations. Next in lemma 16 we establish that total the runtime of Push steps is upper bound by total number of Relabel operations. In lemma 18 we bound the runtime of finding admissible edges. We also argued that if some requests reach at level $\mu + 2$ and matched by Hungarian, the number of such requests are significantly small and thus the total runtime spent at level $\mu + 2$ is $O(n^{2+\delta})$ (Corollary 2). Finally in theorem 3 we prove the bound of the runtime of INCREMENTAL-PUSH-RELABEL . We address the runtime of CREATE-SORTED-EDGE-LIST inside INCREMENTAL-PUSH-RELABEL in theorem 3.

**Lemma 12** *At level $i \leq (\mu + 1)$, for any request $r$, the total number of Relabel operations is upper bounded by $y_i^{\max}$.*

**Proof.** Consider request $r$ enters level $i$. By construction, its dual variable is initialized as $y_r = 0$ upon entering this level. A relabel operation in level $i$ strictly increases $y_r$. In particular, each relabel increments $y_r$ by at least one unit. Since $y_r$ cannot exceed $y_i^{\max}$, the number of relabel operations that can be applied to $r$ within level $i$ is bounded by $y_i^{\max}$.

Moreover, once $r$ exits level $i$ and progresses to a higher level, it cannot return to level $i$. Therefore, no additional relabel operations for $r$ can occur at level $i$. Combining these observations, we conclude that the total number of relabel operations performed on $r$ in level $i$ is at most $y_i^{\max}$. $\qquad \square$

Recall that $B_S^i$ is the set of servers matched at level $i$. Let $\mathcal{R}^i$ be the set of requests such that for any $r \in \mathcal{R}^i$; $level(r) = i$.

**Lemma 13** *After matching $j$-th request, at each level $i \leq (\mu + 1)$, we show that*

$$\left| \sum_{s \in B_S^i} y_i(s) \right| \leq \sum_{r \in \mathcal{R}^i} y_i(r)$$

**Proof.** We establish the lemma using an amortized analysis.

For any server $s \in B_S^i$, define its potential at level $i$ as

$$\psi_i(s) = -y_i(s).$$

By Observation 1, every server with non-zero dual value must belong to the matching $\mathbb{M}_j^i$. Consequently, for any edge $(s, r) \in \mathbb{M}_j^i$ at level $i$, the feasibility condition implies

$$y_i(s) + y_i(r) = \widehat{d}_i(s, r).$$

Since $\widehat{d}_i(s, r) \geq 0$ and $y_i(s) \leq 0$, it follows that

$$\widehat{d}_i(s, r) \leq y_i(r).$$

Moreover, rearranging yields

$$\widehat{d}_i(s, r) - y_i(r) = y_i(s), \quad \text{so that} \quad \psi_i(s) = -y_i(s) = y_i(r) - \widehat{d}_i(s, r).$$

Because all of $\widehat{d}_i(s, r)$, $y_i(r)$, and $\psi_i(s)$ are nonnegative, we further obtain

$$\psi_i(s) \leq y_i(r).$$

Summing over all matched edges at level $i$, we get

$$\sum_{s \in B_S^i} \psi_i(s) \ \leq \ \sum_{r \in \mathcal{R}^i} y_i(r).$$

Since, $\sum_{s \in B_S^i} \psi_i(s) = \left| \sum_{s \in B_S^i} y_i(s) \right|$, we conclude the lemma.

$\qquad \square$

Following Lemma 14 is followed from Agarwal & Sharathkumar (2014) and is crucial to the analysis of runtime. For the sake of completeness we are providing the complete proof.

Consider $\mathbb{M}_n$ be the final matching executed by the algorithm. Let $m_i$ be the matching at level $i$ or above (i.e. $m_i = \cup_{j=i}^{\mu+1} \mathbb{M}_n^i$) and $n_i$ be the number of requests at that level or above, then the following holds.

**Lemma 14** *At any point in the algorithm, at any level $i$, $n_i \leq n^{1-\Phi_i}$, where $\Phi_i = \sum_{k=0}^{i-1} \varphi_k = \frac{3^i - 1}{2}\delta$ and $i \in \{0, \ldots, (\mu + 2)\}$.*

***Proof.*** We claim that

$$\widehat{w}(\mathcal{M}_n^i) \leq \frac{5}{\varepsilon}n^{1-\Phi_i} \tag{8}$$

Suppose this claim is true. Then $n_i \leq n^{1-\Phi_i}$ because $\widehat{w}(\mathcal{M}_n^i) \geq \frac{5}{\varepsilon}n_i$, by Corollary 1(1). Thus it suffices to prove equation 8. we prove it by induction on $i$. For, $i = 0$, $\widehat{d}_0(s,r) - 1 \leq \frac{2n}{\varepsilon\omega}d(s,r)$. Since $\omega \geq w(\mathcal{M}_n^0)/2$ we have,

$$\widehat{w}(\mathcal{M}_n^0) - n \leq \frac{2n}{\varepsilon\omega}w(\mathcal{M}_n^0) \leq \frac{4n}{\varepsilon}$$

This implies $\widehat{w}(\mathcal{M}_n^0) \leq \frac{5n}{\varepsilon}$.

By induction hypothesis, let us assume that

$$\widehat{w}(\mathcal{M}_n^{i-1}) \leq \frac{5}{\varepsilon}n^{1-\Phi_{i-1}}$$

From lemma 10, lemma 11, equation 3 and corollary 1(2), we can write

$$w(\mathcal{M}_n^i, \widehat{d}_{i-1}) \leq \frac{2+\varepsilon}{1-\varepsilon/3}\widehat{w}(\mathcal{M}_n^{i-1}) \leq 2(1+\varepsilon)\frac{5}{\varepsilon}n^{1-\Phi_{i-1}} \tag{9}$$

$w(\mathcal{M}_n^i, \widehat{d}_{i-1})$ denotes the cost of matching $\mathcal{M}_n^i$ under metric $\widehat{d}_{i-1}$. The last inequality in equation 9 follows because $\varepsilon \leq \frac{1}{2}$. On the other hand, by lemma 9(2),

$$(1 - \varepsilon/6)\widehat{d}_i(s,r) \leq \widehat{d}_i(s,r) - 1 \leq \frac{\widehat{d}_{i-1}(s,r)}{2(1+\varepsilon)^2 n^{\varphi_{i-1}}}$$

Therefore,

$$\widehat{d}_i(s,r) \leq \frac{\widehat{d}_{i-1}(s,r)}{2(1+\varepsilon)^2 n^{\varphi_{i-1}}} \tag{10}$$

Combining equation 9 and equation 10, we get,

$$\widehat{w}(\mathcal{M}_n^i) \leq \frac{5}{\varepsilon}n^{1-\Phi_{i-1}-\varphi_{i-1}} = \frac{5}{\varepsilon}n^{1-\Phi_i}$$

$\square$

We bound the total runtime of Relabel operations.

**Lemma 15** *At level $i \leq (\mu + 1)$, total number of relabel operations over all arrivals is upper bounded by $(y_i^{\max} \cdot n^{1-\Phi_i})$.*

***Proof.*** Lemma 12 upper bounds the total number of relabel operations by any request $r$ with $level(r) = i$ by $y_i^{\max}$. Lemma 14 together with Observation 2 implies that at any point of time of execution of INCREMENTAL-PUSH-RELABEL , maximum number of requests at level $i$ does not exceed $n^{1-\Phi_i}$. This conclude the lemma. $\square$

Now we bound the total runtime of Push operations.

**Lemma 16** *At level $i \leq (\mu + 1)$, total number of push operations over all arrivals is upper bounded by $(y_i^{\max} \cdot n^{1-\Phi_i})$.*

**Proof.** Each Push operation consists of single dual decrement of some server. Lemma 13 upper bounds the total dual decrement of servers by total increment of requests matched at that level. However, maximum number of requests at level $i$ does not exceed $n^{1-\Phi_i}$ by Lemma 14 and Observation 2 and the dual of any request upper bounded by $y_i^{\max}$. Hence the lemma. $\square$

We proceed to bound the cost of finding admissible edges. Let us bound the time taken by a single call to FIND-ADMISSIBLE-EDGE .

**Lemma 17** *At level $i \leq (\mu + 1)$, FIND-ADMISSIBLE-EDGE takes $O(n^{1-\Phi_i})$-time to report an admissible edge.*

**Proof.** The runtime of FIND-ADMISSIBLE-EDGE is dominated by the size of the set $|\mathcal{B}_S^i|$. Indeed, checking the admissibility of an edge incident to a free server requires only $O(1)$ time, since the data structure $\mathcal{L}$ is maintained explicitly. Therefore, bounding $|\mathcal{B}_S^i|$ by $O(n^{1-\Phi_i})$ suffices to establish the lemma.

By definition, $\mathcal{B}_S^i$ is the set of servers matched at level $i$ or higher. Note that the number of matched servers is always equal to the number of matched requests. Now we argue the upper bound on matched requests at level $i$. From Observation 2, once a request is matched at or above level $i$, it can never subsequently be matched at a lower level $k < i$. Hence, the number of requests matched at level $i$ or above is monotonically non-decreasing throughout the execution of the algorithm.

Finally, Lemma 14 establishes that, over all arrivals, the total number of requests matched at or above level $i$ is bounded by $O(n^{1-\Phi_i})$. Consequently, the same bound applies to $|\mathcal{B}_S^i|$, which completes the proof. $\square$

In the following lemma, we bound the total cost of finding admissible edges through all requests at level $i$.

**Lemma 18** *At level $i \leq (\mu + 1)$, over all arrivals, total time taken by FIND-ADMISSIBLE-EDGE is $O(\frac{n^{2+\delta}}{\varepsilon})$.*

**Proof.** Lemma 17 shows that any call to FIND-ADMISSIBLE-EDGE requires $O(n^{1-\Phi_i})$ time to find an admissible edge. Each invocation of FIND-ADMISSIBLE-EDGE is immediately followed by either a Push operation or a Relabel operation. Furthermore, Lemma 16 guarantees that the total number of Push operations is upper bounded $(y_i^{\max} \cdot n^{1-\Phi_1})$ and Lemma 15 guarantees that the total number of Relabel operations is upper bounded $(y_i^{\max} \cdot n^{1-\Phi_1})$. Thus, total runtime of finding admissible edges at level $i$ becomes $O(n^{2-2\Phi_i} \cdot y_i^{\max})$. We have, $y_i^{\max} = O\left(\frac{n^{\varphi_i}}{\varepsilon}\right)$. Together, the running time at level $i$ becomes $O(\frac{n^{\varphi_i}}{\varepsilon} \cdot n^{2-2\Phi_i})$.

$$\varphi_i - 2\Phi_i = 3^i \delta - 2\frac{(3^i - 1)}{2}\delta = \delta \tag{11}$$

Hence the runtime becomes $O(\frac{n^{2+\delta}}{\varepsilon})$. $\square$

Now we analysis the total cost spent by INCREMENTAL-PUSH-RELABEL at level $\mu + 2$. At level $\mu + 2$, we resort to the Hungarian algorithm to compute an exact minimum-cost perfect matching for the subgraph induced by the requests that have been promoted to this level. Lemma 14 guarantees that the number of such requests, which we denote by $n_{\mu+2}$, is at most $O(n^\delta)$. The server set at this level, $B_S^{\mu+2} \cup S^{\mathbb{F}}$, has cardinality at most $n$. The Hungarian algorithm, when applied to a bipartite graph with $n_{\mu+2}$ requests and up to $n$ servers, has a time complexity of $O(n_{\mu+2} \cdot n \log n)$ – by using Dijkstra's algorithm with potentials to find each augmenting path. Substituting the bound $n_{\mu+2} = O(n^\delta)$, the cost of processing a single request that reaches this level is $\tilde{O}(n^{1+\delta})$.

Over the entire sequence of $n$ insertions, the total time spent at level $\mu + 2$ is therefore bounded by the number of requests that ever reach this level multiplied by the per-request cost, i.e., $\tilde{O}(n_{\mu+2} \cdot$

$n^{1+\delta}) = \tilde{O}(n^\delta \cdot n^{1+\delta}) = \tilde{O}(n^{1+2\delta})$. For $\delta \leq 1$, this is $\tilde{O}(n^{2+\delta})$, which is within our desired bound. This leads to the following corollary.

**Corollary 2** *Over all arrivals, total time spent by* INCREMENTAL-PUSH-RELABEL *at level* $(\mu + 2)$ *is* $\tilde{O}(n^{2+\delta})$.

**Theorem 3** *The amortized runtime of* INCREMENTAL-PUSH-RELABEL *is* $O(n^{1+\delta} \log^2(\frac{1}{\delta}))$.

***Proof.*** We partition the runtime analysis into two. First we compute the total runtime spent over all levels $i \in \{0, \ldots, (\mu + 1)\}$ over all arrivals. The runtime of the algorithm depends on four major operations. A single invocation to CREATE-SORTED-EDGE-LIST takes no more than $O(n \cdot (\mu + 1))$. The factor of $(\mu + 1)$ appears because computing scaled distance between a pair of points can take $O(\mu + 1)$-time. Thus, over all, CREATE-SORTED-EDGE-LIST costs $O(n^2(\mu + 1))$. Now, we examine the runtime of each of other three operations for fixed level $i$. Following lemma 12, total time of relabeling at level $i$ is $(n^{1-\Phi_i} \cdot y^i_{max})$. Same bound holds for total numbers of push steps following lemma 16. Total time taken by FIND-ADMISSIBLE-EDGE is $O(n^{2-2\Phi_i} \cdot y^{max}_i)$ time. Following lemma 18, this quantity is $O(\frac{n^{2+\delta}}{\varepsilon})$. Hence, the runtime of INCREMENTAL-PUSH-RELABEL at level $i$ is $O(\frac{n^{2+\delta}}{\varepsilon})$. For finding minimum slack, as its runtime is linear to $|\mathcal{B}^i_S|$, the same analysis of runtime of FIND-ADMISSIBLE-EDGE applies. This contributes additional $O(\frac{n^{2+\delta}}{\varepsilon})$ time factor to the runtime of INCREMENTAL-PUSH-RELABEL . Finally we need to update list $\mathcal{L}_r$ for each request $r$, whenever $r$ is matched to a free server. Each time we match a free server, updating this data structure costs $O(1)$ time. Since for each request arrival, we find exactly one free server to match, the cost of this step becomes $O(n)$ over all arrivals.

Recall that, $\varepsilon = \frac{1}{2\log_3(1/\delta)}$. From Corollary 2 and summing over all $O(\log_3(\frac{1}{\delta}))$ many levels, we get total runtime to be $O(n^{2+\delta} \log^2(\frac{1}{\delta}))$.

$\square$

## B  ADDITIONAL EXPERIMENTS

We show our tests and results on the Synthetic data set as described in Section 4.

**Tests:** Similar to NYC-Taxi data, we evaluate four algorithms: Batch Incremental PR, Greedy, QT and Dynamic Euclidean. In this setting, the server set is fixed at $10,000$ and we set $\delta = 0.001$.

**Results:** Batch Incremental PR consistently outperforms Greedy in both cost and runtime. Although QT achieves the best overall runtime performance and Dynamic Euclidean becomes second best, Batch Incremental PR surpasses both QT and Dynamic Euclidean by a significant margin in terms of matching cost (Figure 4(a), Figure 4(b)).

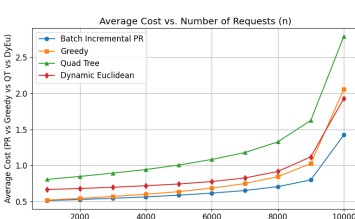
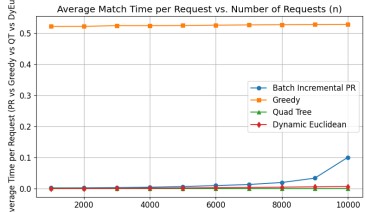

(a) Average cost per request        (b) Average time per request (in seconds)

Figure 4: Plots of Synthetic data

### B.1  $\delta$-SENSITIVITY ANALYSIS

The parameter $\delta$ governs the trade-off between runtime efficiency and solution quality: larger values bias Batch Incremental PR toward achieving lower matching cost, whereas smaller values prioritize

faster execution. To study this trade-off empirically, we evaluate the algorithm on all datasets using $\delta \in \{0.001, 0.01, 0.1\}$. For each setting, we record both the average response time and the resulting average matching cost. The results, summarized in Figures 5, 6, 7, and 8, illustrate how the choice of $\delta$ affects performance across different data characteristics.

Across all datasets, the runtime consistently decreases as $\delta$ becomes smaller: the fastest execution is achieved at $\delta = 0.001$, while $\delta = 0.1$ yields the slowest. The cost trend exhibits the opposite behavior. The largest value, $\delta = 0.1$, consistently produces the lowest cost, whereas $\delta = 0.001$ typically results in the highest cost (or is comparable to $\delta = 0.01$). The intermediate setting, $\delta = 0.01$, usually lies between the two extremes or becomes competitive with $\delta = 0.001$. Overall, these results validate that $\delta$ provides a smooth and predictable mechanism for navigating the runtime–cost spectrum.

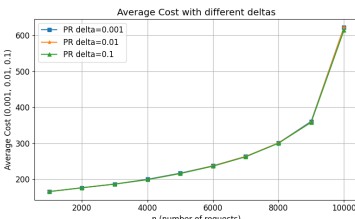
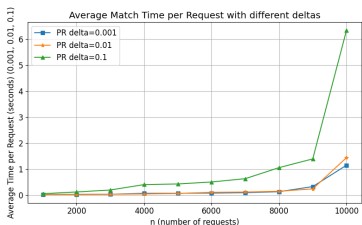

(a) Average cost per request     (b) Average time per request (in seconds)

Figure 5: Cost vs Runtime of Beijing Road Network for $\delta \in \{0.001, 0.01, 0.1\}$

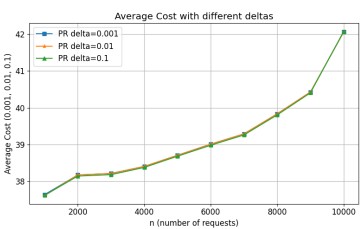
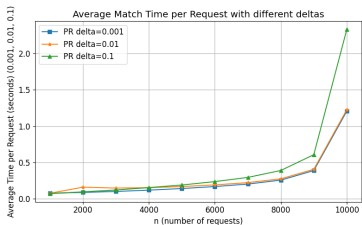

(a) Average cost per request     (b) Average time per request (in seconds)

Figure 6: Cost vs Runtime of MNIST for $\delta \in \{0.001, 0.01, 0.1\}$

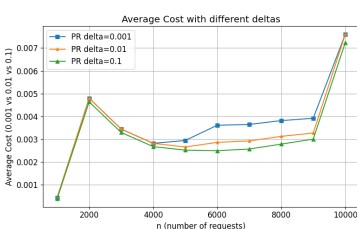
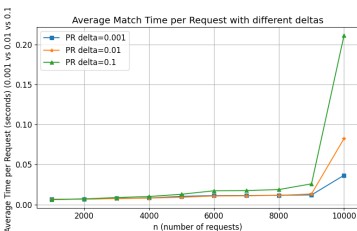

(a) Average cost per request     (b) Average time per request (in seconds)

Figure 7: Cost vs Runtime of NYC-Taxi for $\delta \in \{0.001, 0.01, 0.1\}$

## B.2 EMPIRICAL ANALYSIS OF BATCH-SIZE–RUNTIME TRADEOFFS

Throughout all experiments, we fixed the batch size of Batch Incremental PR to 200. This choice does not necessarily yield the best runtime performance. To illustrate this, we evaluate Batch Incremental PR on the Beijing Road Network using batch sizes $\{100, 200, \ldots, 1000\}$. The results (Figure 9) show that the smallest average matching time is achieved at a batch size of 1000, whereas the largest is observed at a batch size of 100. However, this does not indicate a monotonic improvement with increasing batch size. While the runtime steadily decreases from batch size 100 to 500, it begins to fluctuate for batch sizes between 500 and 1000.

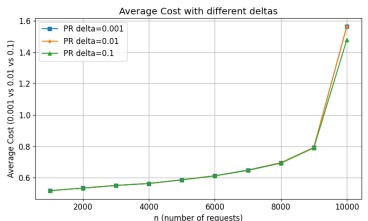 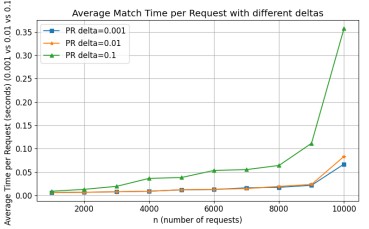

(a) Average cost per request    (b) Average time per request (in seconds)

Figure 8: Cost vs Runtime of Synthetic for $\delta \in \{0.001, 0.01, 0.1\}$

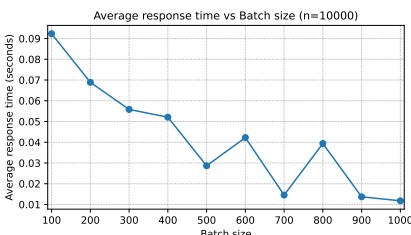

Figure 9: Batchsize-Runtime plot

### B.3 VARIANCE PLOTS

Following we show the variance plots of Batch Incremental PR over different data sets.

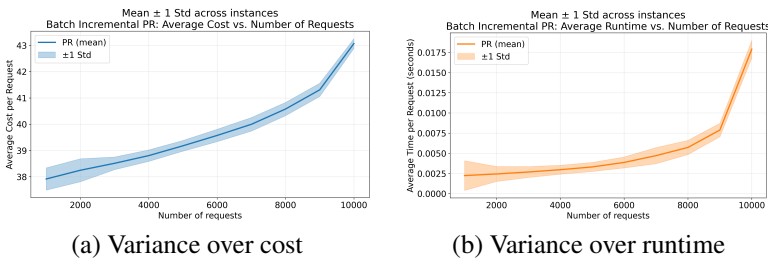

(a) Variance over cost    (b) Variance over runtime

Figure 10: Variance plots on MNIST data

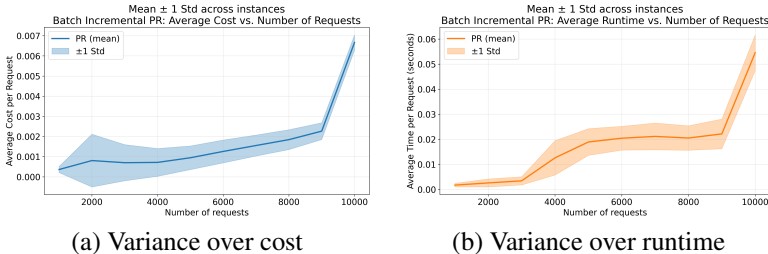

(a) Variance over cost    (b) Variance over runtime

Figure 11: Variance plots of NYC-Taxi data

## C    LLM-USAGE

LLM have only been used to check for typos and grammatical errors.

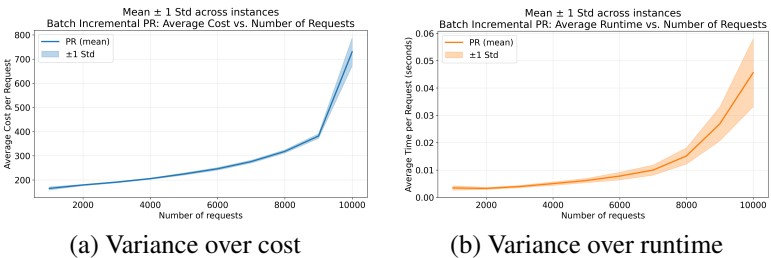

(a) Variance over cost

(b) Variance over runtime

Figure 12: Variance plots on Beijing Road Network

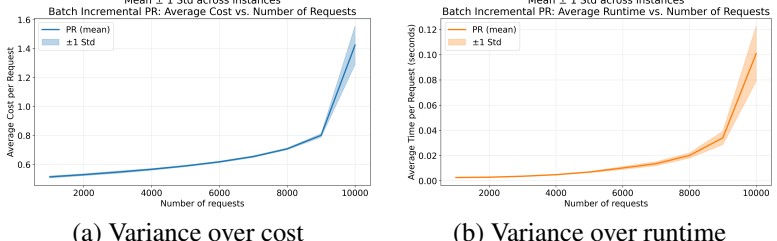

(a) Variance over cost

(b) Variance over runtime

Figure 13: Variance plots of Synthetic data

