# OpenReview forum: "Efficient algorithms for Incremental Metric Bipartite Matching"
_ICLR.cc/2026/Conference — ICLR 2026 Poster_

### Official Review · Reviewer_V2JC · 2025-10-31

**Soundness:** 3
**Presentation:** 2
**Contribution:** 3
**Rating:** 6
**Confidence:** 4

**Summary:**

This paper focuses on the incremental metric bipartite matching problem, where a fixed set of servers S (in a metric space) needs to be dynamically matched to incrementally arriving requests R. The core challenge it addresses is maintaining an approximate minimum-cost matching with efficient update times.

**Strengths:**

It breaks a key barrier of prior incremental matching algorithms by supporting arbitrary metric spaces, which is a critical generalization for applications like high-dimensional ML (e.g., dynamic alignment of MRI scans) and unbalanced logistics. The push-relabel framework is a clever adaptation: replacing augmenting paths with local push/relabel steps reduces per-insertion overhead, making the algorithm feasible for large-scale systems.

The theoretical analysis is comprehensive. The results are not just theoretical, since practical versions are optimized for real-world use, with batch processing that leverages parallelism to handle high-throughput requests.

**Weaknesses:**

The main problem is the experiment design. The authors fix server size and delta. And, logistics scenarios require unbalanced R and S, but in the experiments, |S| = |R|.

One crucial invariant of the algorithm is that a server that is matched at any level 0 ≤ i ≤ µ is only available to requests that are at level i or higher, but this is under-motivated.

**Questions:**

Your guess-and-double trick requires reprocessing all historical requests when ω  is doubled. For large n (e.g., 100k requests), this could add significant runtime. How many such resets occur in practice for your real-world datasets (MNIST, NYC-Taxi)?

Your GPU implementation is mentioned but not detailed. Which submodules (e.g., distance matrix computation, admissible edge search, slack updates) are parallelized? What speedup do you see over the CPU implementation for each submodule, and is a batch size of 200 optimal (or would larger batches improve GPU utilization)?

Logistics scenarios often have bursty request arrivals (e.g., rush hour for taxis). How does your algorithm handle queue buildup under bursty arrivals? Does batch parallelism mitigate this?

---

> ### Author Response · Authors · 2025-11-21
>
> We would like to thank the reviewers for their time and providing valuable comments.
>
> > * “The authors fix server size and delta. And, logistics scenarios require unbalanced $R$ and $S$, but in the experiments, $|S| = |R|$.”
>
> Our algorithm only requires $|S| \geq |R|$. In fact, our runtime and cost guarantees do not assume the two sides are perfectly balanced. In the experiments we fix $|S|$ to $10000$, but $|R| \leq 10000$ at any point of time.
>
> We will add a plot of average response time vs $\delta$ in the final version.
>
> > * “One crucial invariant of the algorithm is that a server that is matched at any level $0 \leq i \leq \mu$ is only available to requests that are at level $i$ or higher, but this is under-motivated.”
>
> Intuitively, the main idea behind our algorithm is as follows - we maintain a hierarchy of scaled metric spaces consisting of $\mu$ many levels. At each level, we maintain a partial matching of the requests that have arrived so far. An important and potentially expensive operation in our push-relabel framework is finding an ‘admissible edge’. This essentially requires a linear scan over the server set. However, by the properties of our dual maintenance, we are able to argue that the size of matched server set drops exponentially as we move up the hierarchy. Further we are able to prove an analogous property for number of requests matched at various levels of hierarchy. Hence, restricting the available set of servers for a free request to its own level or higher allows us to bound the amortized cost of finding an admissible edge. In fact, the argument completely breaks down without the above invariant. Specifically, Lemma 5 uses the monotonicity property crucially.
>
> > * “Your guess-and-double trick requires reprocessing all historical requests when $\omega$ is doubled?”
>
> We are providing the number of times $\omega$ doubled on three different data sets. In all the experiments, the number of servers is fixed at $10000$ while the request set builds up to $10000$.
>
> -  *Beijing City Road Data*: $13$ times.
> -  *MNIST*: $10$ times.
> - *NYC-Taxi*: $7$ times.
>
> > * “Which submodules (e.g., distance matrix computation, admissible edge search, slack updates) are parallelized?”
>
> Although our algorithm is inherently sequential, both key components—push and relabel—lend themselves to parallelization. We implement admissible‐graph construction in parallel, execute the push step using a parallel maximal-matching algorithm on the admissible graph, and parallelize the relabel step by updating entries of the slack matrix concurrently. Our implementation adapts the parallel push–relabel framework of Lahn et al. (https://dl.acm.org/doi/10.5555/3666122.3667070
> ), which provides an $\varepsilon$-additive approximation for optimal transport (and thus min-cost perfect matching) in the static setting.
>
> We have added this discussion to the main body in Section 4 (Experiments).
>
>
> > * “What speedup do you see over the CPU implementation for each submodule”
>
> 1. MNIST
>     - Push $\approx$ $93.996$x
>     - Relabel  $\approx$ $3.731$x
> 2. NYC-Taxi
>     - Push $\approx$ $4.36$x
>     - Relabel $\approx$ $2.95$x
>
> > * “is a batch size of 200 optimal...?”
>
> We tested our algorithm on Beijing City Road Data (Non-Geometric data) with varying batch sizes of $\{100, 200, \ldots, 1000\}$. With a batch size of $100$, the algorithm achieves an average response time of $0.0923$ seconds, while with a batch size of $1000$ it achieves $0.0118$ seconds; however, the plot (Appendix B.1, Figure 9) is not strictly monotonic.
>
> > * “How does your algorithm handle queue buildup under bursty arrivals? Does batch parallelism mitigate this?”
>
> Our algorithm mitigates queue buildup under bursty arrivals in two complementary ways.
>
> 1. **Batch parallelism.**
> In high-throughput periods (e.g., 30-second rush-hour windows), arriving requests can be aggregated into a batch and processed using our GPU-parallel variant. While requests in a batch are not matched simultaneously, the GPU parallelizes the most expensive per-level operations—slack computation and maximal matching in the admissible graph. For datasets like NYC Taxi, 30-second windows typically contain 150–300 arrivals, so our batch size of 200 was chosen to match this range. This batching significantly reduces per-request latency and prevents backlogs.
> 2. **True concurrent processing for streaming arrivals.**
>  Even without batching, the incremental algorithm handles bursty arrivals because each request can begin its push–relabel process *even if earlier requests are still being resolved*. The key property enabling this is that each request operates only on admissible edges with respect to the current duals and matchings at each level. This mitigates queue buildup in streams.
>
> Thus, while batch parallelism gives large throughput gains during predictable bursts, the inherent concurrency of the incremental push–relabel framework prevents bottlenecks even under highly irregular or extreme burstiness.

---

### Official Review · Reviewer_DQGv · 2025-10-31

**Soundness:** 3
**Presentation:** 3
**Contribution:** 4
**Rating:** 8
**Confidence:** 5

**Summary:**

This paper studies the bipartite matching problem for general metrics in the incremental setting. Specifically, given a fixed point set $S$ and an initially empty point set $R$, the goal is to maintain a matching between $S$ and the current $R$ as new points are incrementally inserted into $R$. The paper presents an algorithm that maintains an $O(1/\delta^{0.631})$-approximate minimum-cost matching in amortized time $O(n^{1+\delta})$, ignoring lower-order terms and assuming that the aspect ratio of the underlying metric space is polynomially bounded.

In the static setting, this problem closely relates to estimating the discrete 1-Wasserstein distance, a fundamental quantity in optimization, applied mathematics, machine learning, and theoretical computer science. It has been extensively studied across many computational models. The dynamic (or incremental) setting considered here—where the underlying point sets evolve over time—is both practically motivated and theoretically rich. The only prior work in this area is by Goranci et al.~(ICML'25), which addresses the more general fully dynamic case (supporting both insertions and deletions) but is restricted to low-dimensional Euclidean metrics.

*Technical Contribution*: The central idea builds on the static approximation algorithm of Agarwal and Sharathkumar~(STOC'14) (derived from the classical Gabow--Tarjan framework). However, it is far from obvious how to \emph{dynamize} that algorithm, even in the incremental setting. Skipping many standard details, the paper’s main novelty lies in replacing the traditional *augmenting-path* step with a *push--relabel* step. While individual point insertions may still be expensive in the worst case, the push--relabel mechanism enables strong amortized guarantees. In hindsight, this is a particularly elegant and insightful design choice. This contribution alone, in my view, justifies acceptance at ICLR.

Last but not least, the paper demonstrates that the conceptual simplicity of the proposed approach translates into competitive empirical performance on both real-world and synthetic datasets.

**Strengths:**

-- The paper studies a central problem that lies at the intersection of several fields, and is highly relevant to modern machine learning, especially in the context of emerging computational models.

-- The use of push–relabel techniques to design incremental algorithms, in contrast to the traditional approach of searching for augmenting paths, represents the most novel and technically interesting aspect of the paper.

-- The presentation is generally clear and well-organized, with sufficient intuition and explanation provided beyond the core algorithmic details.

**Weaknesses:**

-- I would expect some better discussion of state-of-the-art resutls on this space (see my comments below:)
-- The result is advertised as applying to general metrics. However, the experiments involve Euclidean data sets. I would expect a bit more effort on this aspect.

**Questions:**

-- The algorithm by Goranci et al. 2025 does work for arbitrary point insertions and deletions, and it's not restricted to pairs of point insertions or deletions (even though it's not explicitly stated); so claiming that as a weakness doesn't seem appropriate.

-- In related works, the paper says that adapting recent breakthrough fast max-flow techniques in the incremental setting seems challenging; however, this has been done: https://arxiv.org/pdf/2311.03174 in the approximate setting (and the paper under review also works in the approximate regime); it still doesn't solve the problem considered in the paper, but some discussion would be due here.

-- Can you explain why you decided to consider only Euclidean data sets in the experiments?

---

> ### Author Response · Authors · 2025-11-21
>
> We would like to thank the reviewers for their time and providing valuable comments.
>
> > * “The algorithm by Goranci et al. 2025 does work for arbitrary point insertions and deletions, and it's not restricted to pairs of point insertions or deletions (even though it's not explicitly stated); so claiming that as a weakness doesn't seem appropriate.”
>
>
> We agree that the techniques of Goranci et al. 2025  could potentially be adapted to settings with one-sided arrivals or departures. But such an extension would require substantial modifications to the proof, which currently relies on balanced instances. We have adjusted the text in the paper accordingly.
>
> > * “In related works, the paper says that adapting recent breakthrough fast max-flow techniques in the incremental setting seems challenging; however, this has been done: https://arxiv.org/pdf/2311.03174 in the approximate setting (and the paper under review also works in the approximate regime); it still doesn't solve the problem considered in the paper, but some discussion would be due here.”
>
>
> We thank the reviewer for pointing out this paper to us. Indeed the authors of this paper give an algorithm to maintain $(1-\varepsilon)$-approximate maximum flow in the incremental edge update setting in $O(m^{o(1)}\varepsilon^{-3})$ update time. Our setting, however, is fundamentally different: we must maintain an approximate min-cost perfect matching under vertex arrivals under metric costs. Extending ideas that maintain approximate flows under incremental updates to instead maintain exact combinatorial structure (such as perfect matchings) while still achieving approximate cost guarantees appears to be an extremely challenging open problem. We have added this discussion along with the reference to the main body of the paper.
>
> > * “Can you explain why you decided to consider only Euclidean data sets in the experiments?”
>
> We ran our experiments on a non-Euclidean real dataset where the metric is the standard shortest path distance. We have updated our paper with the results and can be found in Section 4 (Experiments, Figure 3).

---

### Official Review · Reviewer_HS5c · 2025-11-01

**Soundness:** 3
**Presentation:** 3
**Contribution:** 3
**Rating:** 6
**Confidence:** 4

**Summary:**

The paper considers the \emph{metric bipartite matching problem} in the incremental setting: given two sets of points in a metric space undergoing insertions, the goal is to maintain a minimum-cost matching of one set to the other, where the cost is measured in terms of distances in the metric space, while minimizing the time spent updating the output. The authors present an algorithm achieving an $O(1/\delta^{\sim 0.631})$-approximation to the problem with $\tilde{O}(n^{1+\delta})$ update time.

The only comparable result is that of Goranci et al. (ICML'25), who obtain an $O(1/\epsilon)$-approximation for the problem with $\tilde{O}(n^{O(\epsilon)})$ update time under both point insertions and deletions. However, their algorithm is restricted to low-dimensional Euclidean metrics. Minimum-cost bipartite matching is much more challenging in general metrics, even in the static setting. The result presented by the authors is, to the best of my knowledge, the first algorithm addressing the problem in the dynamic setting for general metrics.

The algorithm presented by the authors closely follows that of Agarwal and Sharathkumar (STOC'14), which achieves the same approximation ratio for the static problem in $O(n^{2+\delta})$ running time (roughly equivalent to the total running time of the algorithm presented by the authors over the entire insertion sequence). The algorithm of Agarwal and Sharathkumar, in turn, builds on the influential work of Gabow and Tarjan (SIAM), who developed a scaling algorithm for exact minimum-cost bipartite perfect matching in graphs, running in $\tilde{O}(n^{2.5})$ time and based on augmenting path elimination techniques. At a high level, Agarwal and Sharathkumar modify this algorithm by approximately completing its scaling steps, resulting in an approximate solution with an almost $n^2$ running time.

The algorithm presented by the authors can be viewed as an incremental implementation of that of Agarwal and Sharathkumar. A crucial challenge in this approach is that the underlying algorithm relies on augmenting path elimination techniques, which are difficult to implement in the dynamic setting, even for maximum cardinality graph matching, when the cost of the optimal solution does not evolve monotonically, as is the case in Euclidean matching. The main technical contribution of the paper lies in overcoming this challenge by formalizing the algorithms of Agarwal and Sharathkumar (and Gabow and Tarjan) as a push-relabel framework, which naturally adapts to the dynamic setting.

The authors also present experiments comparing the performance of their algorithm to greedy and quad-tree-based implementations (Sariel Har-Peled) with respect to Euclidean norms on the MNIST, NYC-TAXI, and synthetic datasets. The results show predictable behaviour in terms of approximation: the algorithm proposed in the paper outperforms both the greedy and quad-tree-based implementations (although it is unclear how it compares to the optimal solution). In terms of update time, the batch-update variant of the proposed algorithm significantly outperforms the greedy approach and slightly underperforms the quad-tree-based implementation.

Overall, I consider the paper to be an important contribution to the study of the widely applied metric minimum-cost bipartite matching problem in the dynamic setting, which has recently received growing attention, due to being the first paper considering the problem in general metrics.

**Strengths:**

The formulation of a variant of the Agarwal and Sharathkumar (and Gabow and Tarjan) algorithm as a push-relabel framework is interesting and, I suspect, will be of independent research interest.

**Weaknesses:**

The algorithm (in contrast to that of Goranci \emph{et al.}) is designed to handle only point insertions. This is somewhat surprising. While for most problems the incremental and decremental settings have proven to be simpler than the fully dynamic one, one might expect this not to be the case for metric minimum-cost matching, as the objective value evolves non-monotonically under partially dynamic updates.

In terms of experimental results, a comparison to the work of Goranci \emph{et al.} would be interesting to see. Based on the theoretical guarantees, with the same slack parameter, one would expect the latter to be more time-efficient at the cost of some loss in approximation ratio relative to the algorithm proposed in this paper. However, since quad-tree-based techniques (such as those of Goranci \emph{et al.}) tend to perform better in practice than in theory, it would be important to see that (at least on low-dimensional Euclidean datasets) the algorithm presented in this paper strictly outperforms the previous approach in at least one aspect.

Furthermore, an experiment not restricted to Euclidean distances (preferably on non-synthetic data) would intuitively should demonstrate a significant improvement over the greedy algorithm in terms of approximation ratio.

**Questions:**

My questions would concern the performance of the algorithm for the test cases described in the weaknesses section.

---

> ### Author Response · Authors · 2025-11-21
>
> We would like to thank the reviewers for their time and providing valuable comments.
>
> > * “In terms of experimental results, a comparison to the work of Goranci \emph{et al.} would be interesting to see.”
>
> We ran a publicly available implementation (https://github.com/Zhengdw/dyn-euc-match) of the algorithm by Goranci et al. on NYC-Taxi and Synthetic datasets. We have updated the paper with results and can be found in Section 4 (Experiments, Figure 3) and Appendix B (Figure 4).
>
>
> > * “an experiment not restricted to Euclidean distances”
>
> We ran our experiments on a non-Euclidean real dataset where the metric is the standard shortest path distance. We have updated our paper with the results and can be found in Section 4 (Experiments, Figure 3).

---

### Official Review · Reviewer_JkD6 · 2025-11-08

**Soundness:** 4
**Presentation:** 4
**Contribution:** 3
**Rating:** 8
**Confidence:** 2

**Summary:**

The paper presents the first streaming and batch-adaptive algorithm maintaining a constant-factor approximation the metric bipartite matching problem. The described algorithm builds upon a previously known algorithm for the offline setting, while making sure that update steps can be executed quickly (when one does amortized time analysis), whereas the worst case is of higher update time. The authors achieve this by utilizing a data structure tailor made for these updates, and by maintaining a collection of partial matchings with a specific property that are a good approximation of the desired matching. Each partial matching corresponds to a different "discretization" of the metric space, at different degrees of refinement. The authors support their theoretical results by empirical experiments.

**Strengths:**

The paper is an important contribution in the field of online-algorithms and metric-driven problems. It is well motivated, and builds upon interesting known ideas in a novel way. The introduction of the "push-relable" subroutine instead of augmenting paths, as well as the careful analysis of the "Find-Admissable-Edge" subroutine allow a clean generalization of the static algorithm. The authors present these deviations well, which contributes to the merit of the paper. The analysis is relatively straightforward, but the ideas are non trivial.

**Weaknesses:**

The paper does not stress in the main body where the assumption of Metricity is necessary, only in the analysis. Since this assumption is critical - a mention is in need.
The experiments need a little more content. The synthetic data analysis seems to be missing (or I have missed it). More importantly, the authors stress that this algorithm is not restricted to the assumption of a Euclidean metric. A comparison with a synthetic dataset that is not Euclidean would be interesting, and will support the strengths of the paper as well.

**Questions:**

Where does the algorithm break if the distance function is not metric?
A couple of typos:
page 4, line 189: $\theta$ should be $\Theta$.
page 5, line 247: $B^i$ should be $B^j$.

---

> ### Author Response · Authors · 2025-11-21
>
> We would like to thank the reviewers for their time and providing valuable comments. The comments regarding typos have been directly incorporated in the updated version of the paper.
>
> > * “Where does the algorithm break if the distance function is not metric?”
>
>
> The algorithm breaks without the metric property because our cost analysis fundamentally relies on the triangle inequality. While one can syntactically construct the same hierarchy of scaled “distances’’ for a non-metric cost function, the guarantees that make the hierarchy useful no longer hold. At each level $i$, the algorithm maintains a partial 1-feasible matching in the scaled metric $\mathcal{M}_i$. A central step in the analysis is the following: if a request remains unmatched at level $i$, the 1-feasibility and dual structure guarantee the existence of a short augmenting path from that request to some server available at level $i$ or above. Because the underlying costs form a metric, the triangle inequality lets us replace this entire augmenting path by a single matching edge whose cost is at most the path length. This is the key argument that allows us to charge all higher-level matching costs to lower-level costs and obtain the global $O(1/\delta^\alpha)$ bound. For non-metric costs, this step fails: a short augmenting path does not imply the existence of a comparably cheap direct edge between its endpoints. As a result, when requests are pushed to higher levels, the matching edges introduced there may incur arbitrarily large distortion relative to lower levels, and the telescoping cost argument used in Lemma 11 (Appendix A.2) collapses. Thus, the approximation guarantee no longer follows.
>
>
> > * Experiments on synthetic and non-Euclidean data
>
> Due to lack of space in the main body, we had added the experimental results on the synthetic dataset in the Appendix B. In addition, we have run a fresh set of experiments on non-Euclidean real dataset. The details of these experiments can be found in Section 4 (Experiments, Figure 3).

---

### Author Response · Authors · 2025-11-21

We thank the reviewers for careful reading of our paper and the helpful comments they provided. One common question is regarding running the experiments on a non-geometric dataset. In order to address this, we have incorporated results on a real life city road network dataset with graph shortest path distances.

---

### Meta-Review · Area_Chair_YSEt · 2025-12-26

**Summary:**

This paper studies min-cost metric bipartite matching in insertion-only setting. An algorithm with poly(1/\delta)-approximation in n^{1 + \delta} update time is given. The result seems to be strong, and the reviews are overall very positive about it. The major concerns is about the lack of non-Euclidean dataset in the experiments. However, it seems new experiments have been added to address the concerns. Overall, this is a solid paper and should be accepted.

**Reviewer Concerns:**

All reviewers questioned the experiment design. New experiment results have been added in the author's rebuttal. I find the new experiments convincing which addresses the reviewers' concerns.

Reviewer DQGv mentioned a reference to which the authors should compare. The authors gave a clear comparison in the rebuttal. The raised reference is tightly related, but its approach cannot be readily applied and does not hurt the novelty of this submission.

No major concerns still remain.

**Reviewer Scores:**

Reviewers with score 8 will mostly keep the score. Reviewer V2JC may raise to 7, because the major concern is about the experiments which is addressed. However, reviewer HS5c will most likely keep the score 6, since the main conservation seems to be on the novelty of the work and the strength of the result.

---

### Decision · Program_Chairs · 2026-01-26

Accept (Poster)